# Beyond Online Balanced Descent: An Optimal Algorithm for Smoothed Online Optimization

**Gautam Goel**[*1] **Yiheng Lin**[*2,1] **Haoyuan Sun**[*1] **Adam Wierman**[1]

[1]California Institute of Technology
[2]Institute for Interdisciplinary Information Sciences, Tsinghua University

## Abstract

We study online convex optimization in a setting where the learner seeks to minimize the sum of a per-round hitting cost and a movement cost which is incurred when changing decisions between rounds. We prove a new lower bound on the competitive ratio of any online algorithm in the setting where the costs are $m$-strongly convex and the movement costs are the squared $\ell_2$ norm. This lower bound shows that no algorithm can achieve a competitive ratio that is $o(m^{-1/2})$ as $m$ tends to zero. No existing algorithms have competitive ratios matching this bound, and we show that the state-of-the-art algorithm, Online Balanced Decent (OBD), has a competitive ratio that is $\Omega(m^{-2/3})$. We additionally propose two new algorithms, Greedy OBD (G-OBD) and Regularized OBD (R-OBD) and prove that both algorithms have an $O(m^{-1/2})$ competitive ratio. The result for G-OBD holds when the hitting costs are quasiconvex and the movement costs are the squared $\ell_2$ norm, while the result for R-OBD holds when the hitting costs are $m$-strongly convex and the movement costs are Bregman Divergences. Further, we show that R-OBD simultaneously achieves constant, dimension-free competitive ratio and sublinear regret when hitting costs are strongly convex.

## 1 Introduction

We consider the problem of Smoothed Online Convex Optimization (SOCO), a variant of online convex optimization (OCO) where the online learner pays a movement cost for changing actions between rounds. More precisely, we consider a game where an online learner plays a series of rounds against an adaptive adversary. In each round, the adversary picks a convex cost function $f_t : \mathbb{R}^d \to \mathbb{R}_{\geq 0}$ and shows it to the learner. After observing the cost function, the learner chooses an action $x_t$ and pays a *hitting cost* $f_t(x_t)$, as well as a *movement cost* $c(x_t, x_{t-1})$, which penalizes the online learner for switching points between rounds.

SOCO was originally proposed in the context of dynamic power management in data centers [28]. Since then it has seen a wealth of applications, from speech animation to management of electric vehicle charging [24–26], and more recently applications in control [21,22] and power systems [5,27]. SOCO has been widely studied in the machine learning community with the special cases of online logistic regression and smoothed online maximum likelihood estimation receiving recent attention [22].

Additionally, SOCO has connections to a number of other important problems in online algorithms and learning. Convex Body Chasing (CBC), introduced in [20], is a special case of SOCO [14]. The

---

Gautam Goel, Yiheng Lin, and Haoyuan Sun contributed equally to this work. This work was supported by NSF grants AitF-1637598 and CNS-1518941, with additional support for Gautam Goel provided by an Amazon AWS AI Fellowship.

problem of designing competitive algorithms for Convex Body Chasing has attracted much recent attention. e.g. [2, 6, 14]. SOCO can also be viewed as a continuous version of the Metrical Task System (MTS) problem (see [9, 11, 12]). A special case of MTS is the celebrated $k-$server problem, first proposed in [30], which has received significant attention in recent years (see [13, 15]).

Given these connections, the design and analysis of algorithms for SOCO and related problems has received considerable attention in the last decade. SOCO was first studied in the scalar setting in [29], which used SOCO to model dynamic "right-sizing" in data centers and gave a 3-competitive algorithm. A 2-competitive algorithm was shown in [8], also in the scalar setting, which matches the lower bound for online algorithms in this setting [1]. Another rich line of work studies how to design competitive algorithms for SOCO when the online algorithm has access to predictions of future cost functions (see [16, 17, 27, 28]).

Despite a large and growing literature on SOCO and related problems, for nearly a decade the only known constant-competitive algorithms that did not use predictions of future costs were for one-dimensional action spaces. In fact, the connections between SOCO and Convex Body Chasing highlight that, in general, one cannot expect dimension-free constant competitive algorithms due to a $\Omega(\sqrt{d})$ lower bound (see [18, 20]). However, recently there has been considerable progress moving beyond the one-dimensional setting for large, important classes of hitting and movement costs.

A breakthrough came in 2017 when [18] proposed a new algorithm, Online Balanced Descent (OBD), and showed that it is constant competitive in all dimensions in the setting where the hitting costs are locally polyhedral and movement costs are the $\ell_2$ norm. The following year, [22] showed that OBD is also constant competitive, specifically $3 + O(1/m)$-competitive, in the setting where the hitting costs are $m$-strongly convex and the movement costs are the squared $\ell_2$ norm. Note that this setting is of particular interest because of its importance for online regression and LQR control (see [22]).

While OBD has proven to be a promising new algorithm, at this point it is not known whether OBD is *optimal* for the competitive ratio, or if there is more room for improvement. This is because there are no non-trivial lower bounds known for important classes of hitting costs, the most prominent of which is the class of strongly convex functions.

**Contributions of this paper.** In this paper we prove the first non-trivial lower bounds on SOCO with strongly convex hitting costs, both for general algorithms and for OBD specifically. These lower bounds show that OBD is not optimal and there is an order-of-magnitude gap between its performance and the general lower bound. Motivated by this gap and the construction of the lower bounds we present two new algorithms, both variations of OBD, which have competitive ratios that match the lower bound. More specifically, we make four main contributions in this paper.

First, we prove a new lower bound on the performance achievable by any online algorithm in the setting where the hitting costs are $m$-strongly convex and the movement costs are the squared $\ell_2$ norm. In particular, in Theorem 1, we show that as $m$ tends to zero, any online algorithm must have competitive ratio at least $\Omega(m^{-1/2})$.

Second, we show that the state-of-the-art algorithm, OBD, cannot match this lower bound. More precisely, in Theorem 2 we show that, as $m$ tends to zero, the competitive ratio of OBD is $\Omega(m^{-2/3})$, an order-of-magnitude higher than the lower bound of $\Omega(m^{-1/2})$. This immediately begs the question: can any online algorithm close the gap and match the lower bound?

Our third contribution answers this question in the affirmative. In Section 4, we propose two novel algorithms, Greedy Online Balanced Descent (G-OBD) and Regularized Online Balanced Descent (R-OBD), which are able to close the gap left open by OBD and match the $\Omega(m^{-1/2})$ lower bound. Both algorithms can be viewed as "aggressive" variants of OBD, in the sense that they chase the minimizers of the hitting costs more aggressively than OBD. In Theorem 3 we show that G-OBD matches the lower bound up to constant factors for quasiconvex hitting costs (a more general class than $m$-strongly convex). In Theorem 4 we show that R-OBD has a competitive ratio that *precisely matches the lower bound, including the constant factors*, and hence can be viewed as an optimal algorithm for SOCO in the setting where the costs are $m$-strongly convex and the movement cost is the squared $\ell_2$ norm. Further, our results for R-OBD hold not only for squared $\ell_2$ movement costs; they also hold for movement costs that are Bregman Divergences, which commonly appear throughout information geometry, probability, and optimization.

Finally, in our last section we move beyond competitive ratio and additionally consider regret. We prove in Theorem 6 that R-OBD can simultaneously achieve bounded, dimension-free competitive ratio and sublinear regret in the case of $m$-strongly convex hitting costs and squared $\ell_2$ movement costs. This result helps close a crucial gap in the literature. Previous work has shown that it not possible for any algorithm to simultaneously achieve both a constant competitive ratio and sublinear regret in general SOCO problems [19]. However, this was shown through the use of linear hitting and movement costs. Thus, the question of whether it is possible to simultaneously achieve a dimension-free, constant competitive ratio and sublinear regret when hitting costs are strongly convex has remained open. The closest previous result is from [18], which showed that OBD can achieve either constant competitive ratio or sublinear regret with locally polyhedral cost functions depending on the "balance condition" used; however both cannot be achieved simultaneously. Our result (Theorem 6), shows that R-OBD can simultaneously provide a constant competitive ratio and sublinear regret for strongly convex cost functions when the movement costs are the squared $\ell_2$ norm.

## 2    Model & Preliminaries

An instance of Smoothed Online Convex Optimization (SOCO) consists of a convex action set $\mathcal{X} \subset \mathbb{R}^d$, an initial point $x_0 \in \mathcal{X}$, a sequence of non-negative convex cost functions $f_1 \ldots f_t : \mathbb{R}^d \to \mathbb{R}_{\geq 0}$, and a movement cost $c : \mathbb{R}^d \times \mathbb{R}^d \to \mathbb{R}_{\geq 0}$. In every round, the environment picks a cost function $f_t$ (potentially adversarially) for an online learner. After observing the cost function, the learner chooses an action $x_t \in \mathbb{R}^d$ and pays a cost that is the sum of the *hitting cost*, $f_t(x_t)$, and the *movement cost*, a.k.a., switching cost, $c(x_t, x_{t-1})$. The goal of the online learner is to minimize its total cost over $T$ rounds: $cost(ALG) = \sum_{t=1}^{T} f_t(x_t) + c(x_t, x_{t-1})$.

We emphasize that it is the movement costs that make this problem interesting and challenging; if there were no movement costs, $c(x_t, x_{t-1}) = 0$, the problem would be trivial, since the learner could always pay the optimal cost simply by picking the action that minimizes the hitting cost in each round, i.e., by setting $x_t = \arg\min_x f_t(x)$. The movement cost couples the cost the learner pays across rounds, which means that the optimal action of the learner depends on unknown future costs.

There is a long literature on SOCO, both focusing on algorithmic questions, e.g., [8, 18, 22, 29], and applications, e.g., [24–26, 28]. The variety of applications studied means that a variety of assumptions about the movement costs have been considered. Motivated by applications to data center capacity management, movement costs have often been taken as the $\ell_1$ norm, i.e., $c(x_1, x_2) = \|x_1 - x_2\|_1$, e.g. [8, 29]. However, recently, more general norms have been considered and the setting of squared $\ell_2$ movement costs has gained attention due to its use in online regression problems and connections to LQR control, among other applications (see [3, 21, 22]).

In this paper, we focus on the setting of the squared $\ell_2$ norm, i.e. $c(x_2, x_1) = \frac{1}{2}\|x_2 - x_1\|_2^2$; however, we also consider a generalization of the $\ell_2$ norm in Section 4.2 where $c$ is the Bregman divergence. Specifically, we consider $c(x_t, x_{t-1}) = D_h(x_t||x_{t-1}) = h(x_t) - h(x_{t-1}) - \langle \nabla h(x_{t-1}), x_t - x_{t-1} \rangle$, where both the potential $h$ and its Fenchel Conjugate $h^*$ are differentiable. Further, we assume that $h$ is $\alpha$-strongly convex and $\beta$-strongly smooth with respect to an underlying norm $\|\cdot\|$. Definitions of each of these properties can be found in the appendix.

Note that the squared $\ell_2$ norm is itself a Bregman divergence, with $\alpha = \beta = 1$ and $\|\cdot\| = \|\cdot\|_2$, $D_h(x_t||x_{t-1}) = \frac{1}{2}\|x_t - x_{t-1}\|_2^2$. However, more generally, when $h(y) = \sum_i y_i \ln y_i$ with domain $\Delta_n = \{y \in [0,1]^n \mid \sum_i y_i = 1\}$, $D_h(x_t||x_{t-1})$ is the Kullback-Liebler divergence (see [7]). Further, $h$ is $\frac{1}{2\ln 2}$-strongly convex and $\frac{1}{\delta \ln 2}$-strongly smooth in the domain $\mathcal{X} = P_\delta = \{y \in [0,1]^n \mid \sum_i y_i = 1, y_i \geq \delta\}$ (see [18]). This extension is important given the role Bregman divergence plays across optimization and information theory, e.g., see [4, 31].

Like for movement costs, a variety of assumptions have been made about hitting costs. In particular, because of the emergence of pessimistic lower bounds when general convex hitting costs are considered, papers typically have considered restricted classes of functions, e.g., locally polyhedral [18] and strongly convex [22]. In this paper, we focus on hitting costs that are $m$-strongly convex; however our results in Section 4.1 generalize to the case of quasiconvex functions.

**Competitive Ratio and Regret.** The primary goal of the SOCO literature is to design online algorithms that (nearly) match the performance of the offline optimal algorithm. The performance metric used to evaluate an algorithm is typically the *competitive ratio* because the goal is to learn in

---

**Algorithm 1** Online Balanced Descent (OBD)

---
1: **procedure** OBD($f_t, x_{t-1}, \gamma$)                                   ▷ Procedure to select $x_t$
2:     $v_t \leftarrow \arg\min_x f_t(x)$
3:     Let $x(l) = \prod_{K_t^l}(x_{t-1})$. Initialize $l = f_t(v_t)$. Here $K_t^l = \{x | f_t(x) \le l\}$.
4:     Increase $l$. Stop when $c(x(l), x_{t-1}) = \gamma(l - f_t(v_t))$.
5:     $x_t \leftarrow x(l)$.
6:     **return** $x_t$

---

an environment that is changing dynamically and is potentially adversarial. The competitive ratio is the worst-case ratio of total cost incurred by the online learner and the offline optimal costs. The cost of the offline optimal is defined as the minimal cost of an algorithm if it has full knowledge of the sequence of costs $\{f_t\}$, i.e. $cost(OPT) = \min_{x_1 \dots x_T} \sum_{t=1}^T f_t(x_t) + c(x_t, x_{t-1})$. Using this, the *competitive ratio* is defined as $\sup_{f_1 \dots f_T} cost(ALG)/cost(OPT)$.

Note that another important performance measure of interest is the *regret*. In this paper, we study a generalization of the classical regret called the $L$-constrained regret, which is defined as follows. The $L$-*(constrained) dynamic regret* of an online algorithm $ALG$ is $\rho_L(T)$ if for all sequences of cost functions $f_t, \cdots, f_T$, we have $cost(ALG) - cost(OPT(L)) \le \rho_L(T)$ where $OPT(L)$ is the cost of an $L$-constrained offline optimal solution, i.e., one with movement cost upper bounded by $L$: $OPT(L) = \min_{x \in \mathcal{X}^T} \sum_{t=1}^T f_t(x_t) + c(x_t, x_{t-1})$ subject to $\sum_{t=1}^T c(x_t, x_{t-1}) \le L$.

As the definitions above highlight, the regret and competitive ratio both compare with the cost of an offline optimal solution, however regret constrains the movement allowed by the offline optimal. The classical notion of regret focuses on the static optimal ($L = 0$), but relaxing that to allow limited movement bridges regret and the competitive ratio since, as $L$ grows, the $L$-constrained offline optimal approaches the offline (dynamic) optimal. Intuitively, one can think of regret as being suited for evaluating learning algorithms in (nearly) static settings while the competitive ratio as being suited for evaluating learning algorithms in dynamic settings.

**Online Balanced Descent.** The state-of-the-art algorithm for SOCO is Online Balanced Descent (OBD). OBD, which is formally defined in Algorithm 1, uses the operator $\Pi_K(x) : \mathbb{R}^d \to K$ to denote the $\ell_2$ projection of $x$ onto a convex set $K$; and this operator is defined as $\Pi_K(x) = \arg\min_{y \in K} \|y - x\|_2$. Intuitively, it works as follows. In every round, OBD projects the previously chosen point $x_{t-1}$ onto a carefully chosen level set of the current cost function $f_t$. The level set is chosen so that the hitting costs and movement costs are "balanced": in every round, the movement cost is at most a constant $\gamma$ times the hitting cost. The balance helps ensure that the online learner is matching the offline costs. Since neither cost is too high, OBD ensures that both are comparable to the offline optimal. The parameter $\gamma$ can be tuned to give the optimal competitive ratio and the appropriate level set can be efficiently selected via binary search.

Implicitly, OBD can be viewed as a proximal algorithm with a dynamic step size [32], in the sense that, like proximal algorithms, OBD iteratively projects the previously chosen point onto a level set of the cost function. Unlike traditional proximal algorithms, OBD considers several different level sets, and carefully selects the level set in every round so as to balance the hitting and movement costs. We exploit this connection heavily when designing Regularized OBD (R-OBD), which is a proximal algorithm with a special regularization term added to the objective to help steer the online learner towards the hitting cost minimizer in each round.

OBD was proposed in [18], where the authors show that it has a constant, dimension-free competitive ratio in the setting where the movement costs are the $\ell_2$ norm and the hitting costs are locally polyhedral, i.e. grow at least linearly away from the minimizer. This was the first time an algorithm had been shown to be constant competitive beyond one-dimensional action spaces. In the same paper, a variation of OBD that uses a different balance condition was proven to have $O(\sqrt{TL})$ $L$-constrained regret for locally polyhedral hitting costs. OBD has since been shown to also have a constant, dimension-free competitive ratio when movement costs are the squared $\ell_2$ norm and hitting costs are strongly convex, which is the setting we consider in this paper. However, up until this paper, lower bounds for the strongly convex setting did not exist and it was not known whether the performance of OBD in this setting is optimal or if OBD can simultaneously achieve sublinear regret and a constant, dimension-free competitive ratio.

# 3 Lower Bounds

Our first set of results focuses on lower bounding the competitive ratio achievable by online algorithms for SOCO. While [18] proves a general lower bound for SOCO showing that the competitive ratio of any online algorithm is $\Omega(\sqrt{d})$, where $d$ is the dimension of the action space, there are large classes of important problems where better performance is possible. In particular, when the hitting costs are $m$-strongly convex, [22] has shown that OBD provides a dimension-free competitive ratio of $3 + O(1/m)$. However, no non-trivial lower bounds are known for the strongly convex setting.

Our first result in this section shows a general lower bound on the competitive ratio of SOCO algorithms when the hitting costs are strongly convex and the movement costs are quadratic. Importantly, there is a gap between this bound and the competitive ratio for OBD proven in [22]. Our second result further explores this gap. We show a lower bound on the competitive ratio of OBD which highlights that OBD cannot achieve a competitive ratio that matches the general lower bound. This gap, and the construction used to show it, motivate us to propose new variations of OBD in the next section. We then prove that these new algorithms have competitive ratios that match the lower bound.

We begin by stating the first lower bound for strongly convex hitting costs in SOCO.

**Theorem 1.** *Consider hitting cost functions that are $m$-strongly convex with respect to $\ell_2$ norm and movement costs given by $\frac{1}{2} \|x_t - x_{t-1}\|_2^2$. Any online algorithm must have a competitive ratio at least $\frac{1}{2} \left( 1 + \sqrt{1 + \frac{4}{m}} \right)$.*

Theorem 1 is proven in the appendix using an argument that leverages the fact that, when the movement cost is quadratic, reaching a target point via one large step is more costly than reaching it by taking many small steps. More concretely, to prove the lower bound we consider a scenario on the real line where the online algorithm encounters a sequence of cost functions whose minimizers are at zero followed by a very steep cost function whose minimizer is at $x = 1$. Without knowledge of the future, the algorithm has no incentive to move away from zero until the last step, when it is forced to incur a large cost; however, the offline adversary, with full knowledge of the cost sequence, can divide the journey into multiple small steps.

Importantly, the lower bound in Theorem 1 highlights the dependence of the competitive ratio on $m$, the convexity parameter. It shows that the case where online algorithms do the worst is when $m$ is small, and that algorithms that match the lower bound up to a constant are those for which the competitive ratio is $O(m^{-1/2})$ as $m \to 0^+$. Note that our results in Section 4 show that there exists online algorithms that precisely achieve the competitive ratio in Theorem 1. However, in contrast, the following shows that OBD cannot match the lower bound in Theorem 1.

**Theorem 2.** *Consider hitting cost functions that are $m$-strongly convex with respect to $\ell_2$ norm and a movement costs given by $\frac{1}{2} \|x_t - x_{t-1}\|_2^2$. The competitive ratio of OBD is $\Omega(m^{-\frac{2}{3}})$ as $m \to 0^+$, for any fixed balance parameter $\gamma$.*

As we have discussed, OBD is the state-of-the-art algorithm for SOCO, and has been shown to provide a competitive ratio of $3 + O(1/m)$ [22]. However, Theorem 2 highlights a gap between OBD and the general lower bound. If the lower bound is achievable (which we prove it is in the next section), this implies that OBD is a sub-optimal algorithm.

The proof of Theorem 2 gives important intuition about what goes wrong with OBD and how the algorithm can be improved. Specifically, our proof of Theorem 2 considers a scenario where the cost functions have minimizers very near each other, but OBD takes a series of steps without approaching the minimizing points. The optimal is able to pay little cost and stay near the minimizers, but OBD never moves enough to be close to the minimizers. Figure 1 illustrates the construction, showing OBD moving along the circumference of a circle, while the offline optimal stays near the origin.

# 4 Algorithms

The lower bounds in Theorem 1 and Theorem 2 suggest a gap between the competitive ratio of OBD and what is achievable via an online algorithm. Further, the construction used in the proof of Theorem 2 highlights the core issue that leads to inefficiency in OBD. In the construction, OBD takes a large step from $x_{t-1}$ to $x_t$, but the offline optimal, $x_t^*$, only decreases by a very small amount. This means

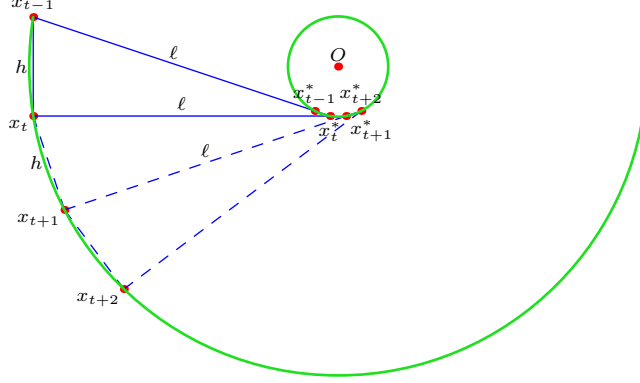

Figure 1: *Counterexample used to prove Theorem 2. In the figure, $\{x_t\}$ are the choices of OBD and $\{x_t^*\}$ are the choices of the offline optimal.*

that OBD is continually chasing the offline optimal but never closing the gap. In this section, we take inspiration from this example and develop two new algorithms that build on OBD but ensure that the gap to the offline optimal $x_t^*$ shrinks.

How to ensure that the gap to the offline optimal shrinks is not obvious since, without the knowledge about the future, it is impossible to determine how $x_t^*$ will evolve. A natural idea is to determine an online estimate of $x_t^*$ and then move towards that estimate. Motivated by the construction in the proof of Theorem 2, we use the minimizer of the hitting cost at round $t$, $v_t$, as a rough estimate of the offline optimal and ensure that we close the gap to $v_t$ in each round.

There are a number of ways of implementing the goal of ensuring that OBD more aggressively moves toward the minimizer of the hitting cost each round. In this section, we consider two concrete approaches, each of which (nearly) matches the lower bound in Theorem 1.

The first approach, which we term Greedy OBD (Algorithm 2) is a two-stage algorithm, where the first stage applies OBD and then a second stage explicitly takes a step directly towards the minimizer (of carefully chosen size). We introduce the algorithm and analyze its performance in Section 4.1. Greedy OBD is order-optimal, i.e. matches the lower bound up to constant factors, in the setting of squared $\ell_2$ norm movement costs and quasiconvex hitting costs.

The second approach for ensuring that OBD moves aggressively toward the minimizer uses a different view of OBD. In particular, Greedy OBD uses a *geometric view* of OBD, which is the way OBD has been presented previously in the literature. Our second view uses a "local view" of OBD that parallels the *local view* of gradient descent and mirror descent, e.g., see [7, 23]. In particular, the choice of an action in OBD can be viewed as the solution to a per-round *local* optimization. Given this view, we ensure that OBD more aggressively tracks the minimizer by adding a regularization term to this local optimization which penalizes points which are far from the minimizer. We term this approach Regularized OBD (Algorithm 3), and study it in Section 4.2. Note that Regularized OBD has a competitive ratio that precisely matches the lower bound, including the constant factors, when movement costs are Bregman divergences and hitting costs are $m$-strongly convex. Thus, it applies for more general movement costs than Greedy OBD but less general hitting costs.

## 4.1 Greedy OBD

The formal description of Greedy Online Balanced Descent (G-OBD) is given in Algorithm 2. G-OBD has two steps each round. First, the algorithm takes a standard OBD step from the previous point $x_{t-1}$ to a new point $x_t'$, which is the projection of $x_{t-1}$ onto a level set of the current hitting cost $f_t$, where the level set is chosen to balance hitting and movement costs. G-OBD then takes an additional step directly towards the minimizer of the hitting cost, $v_t$, with the size of the step chosen based on the convexity parameter $m$. G-OBD can be implemented efficiently using the same approach as described for OBD [18]. G-OBD has two parameters $\gamma$ and $\mu$. The first, $\gamma$, is the balance parameter in OBD and the second, $\mu$, is a parameter controlling the size of the step towards the

---
**Algorithm 2** Greedy Online Balanced Descent (G-OBD)
---
1: **procedure** G-OBD($f_t, x_{t-1}$)                                  ▷ Procedure to select $x_t$
2:     $v_t \leftarrow \arg\min_x f_t(x)$
3:     $x'_t \leftarrow OBD(f_t, x_{t-1}, \gamma)$
4:     **if** $\mu\sqrt{m} \geq 1$ **then**
5:         $x_t \leftarrow v_t$
6:     **else**
7:         $x_t \leftarrow \mu\sqrt{m}v_t + (1 - \mu\sqrt{m})x'_t$
8:     **return** $x_t$
---

---
**Algorithm 3** Regularized OBD (R-OBD)
---
1: **procedure** R-OBD($f_t, x_{t-1}$)                                  ▷ Procedure to select $x_t$
2:     $v_t \leftarrow \arg\min_x f_t(x)$
3:     $x_t \leftarrow \arg\min_x f_t(x) + \lambda_1 c(x, x_{t-1}) + \lambda_2 c(x, v_t)$
4:     **return** $x_t$
---

minimizer $v_t$. Note that the two-step approach of G-OBD is reminiscent of the two-stage algorithm used in [10]; however the resulting algorithms are quite distinct.

While the addition of a second step in G-OBD may seem like a small change, it improves performance by an order-of-magnitude. We prove that G-OBD asymptotically matches the lower bound proven in Theorem 2 not just for $m$-strongly convex hitting costs, but more broadly to quasiconvex costs.

**Theorem 3.** *Consider quasiconvex hitting costs such that $f_t(x) \geq f_t(v_t) + \frac{m}{2} \|x - v_t\|_2^2$ and movement costs $c(x_t, x_{t-1}) = \frac{1}{2} \|x_t - x_{t-1}\|_2^2$. G-OBD with $\gamma = 1, \mu = 1$ is an $O\left(m^{-1/2}\right)$-competitive algorithm as $m \to 0^+$.*

## 4.2 Regularized OBD

The G-OBD framework is based on the geometric view of OBD used previously in literature. There are, however, two limitations to this approach. First, the competitive ratio obtained, while having optimal asymptotic dependence on $m$, does not not match the constants in the lower bound of Theorem 1. Second, G-OBD requires repeated projections, which makes efficient implementation challenging when the functions $f_t$ have complex geometry.

Here, we present a variation of OBD based on a *local view* that overcomes these limitations. Regularized OBD (R-OBD) is computationally simpler and provides a competitive ratio that matches the constant factors in the lower bound in Theorem 1. However, unlike G-OBD, our analysis of R-OBD does not apply to quasiconvex hitting costs. R-OBD is described formally in Algorithm 3. In each round, R-OBD picks a point that minimizes a weighted sum of the hitting and movement costs, as well as a regularization term which encourages the algorithm to pick points close to the minimizer of the current hitting cost function, $v_t = \arg\min_x f_t(x)$. Thus, R-OBD can be implemented efficiently using two invocations of a convex solver. Note that R-OBD has two parameters $\lambda_1$ and $\lambda_2$ which adjust the weights of the movement cost and regularizer respectively.

While it may not be immediately clear how R-OBD connects to OBD, it is straightforward to illustrate the connection in the squared $\ell_2$ setting. In this case, computing $x_t = \arg\min_x f_t(x) + \frac{\lambda_1}{2} \|x - x_{t-1}\|_2^2$ is equivalent to doing a projection onto a level set of $f_t$, since the selection of the minimizer can be restated as the solution to $\nabla f_t(x_t) + \lambda_1(x_t - x_{t-1}) = 0$. Thus, without the regularizer, the optimization in R-OBD gives a local view of OBD and then the regularizer provides more aggressive movement toward the minimizer of the hitting cost.

Not only does the local view lead to a computationally simpler algorithm, but we prove that R-OBD matches the constant factors in Theorem 1 precisely, not just asymptotically. Further, it does this not just in the setting where movement costs are the squared $\ell_2$ norm, but also in the case where movement costs are Bregman divergences.

**Theorem 4.** *Consider hitting costs that are $m-$strongly convex with respect to a norm $\|\cdot\|$ and movement costs defined as $c(x_t, x_{t-1}) = D_h(x_t \| x_{t-1})$, where $h$ is $\alpha$-strongly convex and $\beta$-strongly*

*smooth with respect to the same norm. Additionally, assume $\{f_t\}, h$ and its Fenchel Conjugate $h^*$ are differentiable. Then, R-OBD with parameters $1 \geq \lambda_1 > 0$ and $\lambda_2 \geq 0$ has a competitive ratio of $\max\left(\frac{m+\lambda_2\beta}{\lambda_1} \cdot \frac{1}{m}, 1 + \frac{\beta^2}{\alpha} \cdot \frac{\lambda_1}{\lambda_2\beta+m}\right)$. If $\lambda_1$ and $\lambda_2$ satisfy $m + \lambda_2\beta = \frac{\lambda_1 m}{2}\left(1 + \sqrt{1 + \frac{4\beta^2}{\alpha m}}\right)$ then the competitive ratio is $\frac{1}{2}\left(1 + \sqrt{1 + \frac{4\beta^2}{m\alpha}}\right)$.*

Theorem 4 focuses on movement costs that are Bregman divergences, which generalizes the case of squared $\ell_2$ movement costs. To recover the squared $\ell_2$ case, we use $\|\cdot\| = \|\cdot\|_2$ and $\alpha = \beta = 1$, which results in a competitive ratio of $\frac{1}{2}(1 + \sqrt{1 + 4/m})$. This competitive ratio matches exactly with the lower bound claimed in Theorem 1. Further, in this case the assumption in Theorem 4 that the hitting cost functions are differentiable is not required (see Theorem **??** in the appendix).

It is also interesting to investigate the settings of $\lambda_1$ and $\lambda_2$ that yield the optimal competitive ratio. Setting $\lambda_2 = 0$ achieves the optimal competitive ratio as long as $\lambda_1 = 2\left(1 + \sqrt{1 + \frac{4\beta^2}{\alpha m}}\right)^{-1}$. By restating the update rule in R-OBD as $\nabla f_t(x_t) = \lambda_1(\nabla h(x_{t-1}) - \nabla h(x_t))$, we see that R-OBD with $\lambda_2 = 0$ can be interpreted as "one step lookahead mirror descent". Further R-OBD with $\lambda_2 = 0$ can be implemented even when we do not know the location of the minimizer $v_t$. For example, when $h(x) = \frac{1}{2}\|x\|_2^2$, we can run gradient descent starting at $x_{t-1}$ to minimize the strongly convex function $f_t(x) + \frac{\lambda_1}{2}\|x - x_{t-1}\|_2^2$. Only local gradients will be queried in this process. However, the following lower bound highlights that this simple form comes at some cost in terms of generality when compared with our results for G-OBD.

**Theorem 5.** *Consider quasiconvex hitting costs such that $f_t(x) - f_t(v_t) \geq \frac{m}{2}\|x - v_t\|_2^2$ and movement costs given by $c(x_t, x_{t-1}) = \frac{1}{2}\|x_t - x_{t-1}\|_2^2$. Regularized OBD has a competitive ratio of $\Omega(1/m)$ when $\lambda_2 = 0$.*

## 5  Balancing Regret and Competitive Ratio

In the previous sections we have focused on the competitive ratio; however another important performance measure is *regret*. In this section, we consider the $L$-constrained dynamic regret. The motivation for our study is [19], which provides an impossibility result showing that no algorithm can simultaneously maintain a constant competitive ratio and a sub-linear regret in the general setting of SOCO. However, [19] utilizes linear hitting costs in its construction and thus it is an open question as to whether this impossibility result holds for strongly convex hitting costs. In this section, we show that the impossibility result does not hold for strongly convex hitting costs. To show this, we first characterize the parameters for which R-OBD gives sublinear regret.

**Theorem 6.** *Consider hitting costs that are $m-$strongly convex with respect to a norm $\|\cdot\|$ and movement costs defined as $c(x_t, x_{t-1}) = D_h(x_t\|x_{t-1})$, where $h$ is $\alpha$-strongly convex and $\beta$-strongly smooth with respect to the same norm. Additionally, assume $\{f_t\}, h$ and its Fenchel Conjugate $h^*$ are differentiable. Further, suppose that $\|\nabla h(x)\|_*$ is bounded above by $G < \infty$, the diameter of the feasible set $\mathcal{X}$ is bounded above by $D$, and $\nabla h(0) = 0$. Then, for $\lambda_1, \lambda_2$ such that $\lambda_1 \geq 1 - \frac{m}{4\beta}$ and $\lambda_2 = \eta(T, L, D, G)$, where $\eta(T, L, D, G)$ is such that $\lim_{T\to\infty} \eta(T, L, D, G) \cdot \frac{D^2}{G}\sqrt{\frac{T}{L}} < \infty$, the $L$-constrained regret of R-OBD is $O(G\sqrt{TL})$.*

Theorem 6 highlights that $O(G\sqrt{TL})$ regret can be achieved when $\lambda_1 \geq 1 - \frac{m}{4\beta}$ and $\lambda_2 \leq \frac{KG}{D^2} \cdot \sqrt{\frac{L}{T}}$ for some constant $K$. This suggests that the tendency to aggressively move towards the minimizer should shrink over time in order to achieve a small regret. It is not possible to use Theorem 6 to simultaneously achieve the optimal competitive ratio and $O(G\sqrt{TL})$ regret for all strongly convex hitting costs ($m > 0$). However, the corollary below shows that it is possible to simultaneously achieve a dimension-free, constant competitive ratio and an $O(G\sqrt{TL})$ regret for all $m > 0$. An interesting open question that remains is whether it is possible to develop an algorithm that has sublinear regret and matches the optimal order for competitive ratio.

**Corollary 1.** *Consider the same conditions as in Theorem 6 and fix $m > 0$. R-OBD with parameters $\lambda_1 = \max\left(2\left(1 + \sqrt{1 + \frac{4\beta^2}{\alpha m}}\right)^{-1}, 1 - \frac{m}{4\beta}\right), \lambda_2 = 0$ has an $O(G\sqrt{TL})$ regret and is $\max\left(\frac{1}{2}\left(1 + \sqrt{1 + \frac{4\beta^2}{\alpha m}}\right), 1 - \frac{\beta}{4\alpha} + \frac{\beta^2}{\alpha m}\right)$-competitive.*

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
