[Supplementary Material]

# Appendices

The appendices that follow provide the proofs of the results in the body of the paper. Throughout the proofs in the appendix we use the following notation to denote the hitting and movement costs of the online learner: $H_t := f_t(x_t)$ and $M_t := c(x_t, x_{t-1})$, where $x_t$ is the point chosen by the online algorithm at time $t$. Similarly, we denote the hitting and movement costs of the offline optimal (adversary) as $H_t^* := f_t(x_t^*)$ and $M_t^* := c(x_t^*, x_{t-1}^*)$, where $x_t^*$ is the point chosen by the offline optimal at time $t$.

Before moving to the proofs, we summarize a few standard definitions that are used throughout the paper.

**Definition 1.** *A function $f : \mathcal{X} \to \mathbb{R}$ is $\alpha$-strongly convex with respect to a norm $\|\cdot\|$ if for all $x, y$ in the relative interior of the domain of $f$ and $\lambda \in (0, 1)$, we have*

$$f(\lambda x + (1 - \lambda)y) \le \lambda f(x) + (1 - \lambda)f(y) - \frac{\alpha}{2}\lambda(1 - \lambda)\|x - y\|^2.$$

**Definition 2.** *A function $f : \mathcal{X} \to \mathbb{R}$ is $\beta$-strongly smooth with respect to a norm $\|\cdot\|$ if $f$ is everywhere differentiable and if for all $x, y$ we have*

$$f(y) \le f(x) + \langle \nabla f(x), y - x \rangle + \frac{\beta}{2}\|y - x\|^2.$$

**Definition 3.** *A function $f : \mathbb{R}^d \to \mathbb{R}$ is quasiconvex if its domain $\mathcal{X}$ and all its sublevel sets*

$$S_\alpha = \{x \in \mathcal{X} \mid f(x) \le \alpha\},$$

*for $\alpha \in \mathbb{R}$, is convex.*

**Definition 4.** *For a norm $\|\cdot\|$ in $\mathcal{X}$, its dual norm (on $\mathcal{X}$) $\|\cdot\|_*$ is defined to be*

$$\|y\|_* = \sup\{\langle x, y \rangle \mid \|x\| \le 1\}.$$

**Definition 5.** *For a convex function $f : \mathcal{X} \to \mathbb{R}$, its Fenchel Conjugate $f^*$ is defined to be*

$$f^*(y) = \sup\{\langle x, y \rangle - f(x) \mid x \in \mathcal{X}\}.$$

Next, we introduce a few technical lemmas that are important throughout our analysis.

The first technical lemma is a characterization of strongly convex functions.

**Lemma 1.** *Suppose $f$ is $\alpha$-strongly convex for some $\alpha > 0$ with respect to some norm $\|\cdot\|$ and both $f$ and $f^*$ are differentiable, then the first condition implies the second condition and the third condition:*

    *1. $\forall x, y, f(y) \le f(x) + \langle \nabla f(x), y - x \rangle + \frac{\beta}{2}\|x - y\|^2$;*

    *2. $\forall x, y, f(y) \ge f(x) + \langle \nabla f(x), y - x \rangle + \frac{1}{2\beta}\|\nabla f(x) - \nabla f(y)\|_*^2$;*

    *3. $\forall x, y, \|\nabla f(x) - \nabla f(y)\|_* \le \beta \|x - y\|$.*

To prove Lemma 1, we use Lemma 2, Lemma 3, and Lemma 4 below.

The following lemma is Theorem 6 in [25].

**Lemma 2.** *If $f$ is convex and closed, the following two conditions are equivalent:*

    *1. $\forall x, y, f(y) \ge f(x) + \langle \nabla f(x), y - x \rangle + \frac{\beta}{2}\|x - y\|^2$;*

    *2. $\forall x, y, f^*(y) \le f^*(x) + \langle \nabla f^*(x), y - x \rangle + \frac{1}{2\beta}\|x - y\|_*^2$*

*i.e. $f$ is $\beta$-strongly convex w.r.t some norm $\|\cdot\|$ if and only if $f^*$ is $\frac{1}{\beta}$-strongly smooth w.r.t the dual norm $\|\cdot\|_*$.*

The next lemma is a special case of Lemma 17 in [34].

472 **Lemma 3.** *Let f be a closed, convex, and differentiable function. Then we have*
$$f^*(\nabla f(x)) + f(x) = \langle \nabla f(x), x \rangle.$$

473 Now we prove a technical result that describes a property of the gradient of the Fenchel Conjugate.

474 **Lemma 4.** *Suppose f is $\alpha-$strongly convex for some $\alpha > 0$ with respect to some norm $\|\cdot\|$ and both*
475 *f and $f^*$ are differentiable. Then we have*
$$x = \nabla f^* \left( \nabla f(x) \right), \forall x.$$

476 *Proof.* For convenience, we define $y = \nabla f(x)$ and $x' = \nabla f^*(y)$. It suffices to prove that $x' = x$.

477 By Lemma 3, we obtain
$$f^*(y) + f(x) = \langle y, x \rangle = \langle x, y \rangle. \tag{1}$$

478 Again by Lemma 3, we have

$$f(x') + f^*(y) = f^{**}(x') + f^*(y) = \langle x', y \rangle, \tag{2}$$
479 where we use the fact that $f^{**} = f$.

480 Combining inequalities (1) and (2), we obtain
$$0 = f(x) - f(x') - \langle x - x', y \rangle = f(x) + \langle x' - x, \nabla f(x) \rangle - f(x') \le -\frac{\alpha}{2} \|x - x'\|^2,$$
481 where in the last inequality we use the definition of $\alpha-$strongly convex. Therefore we have proved
482 that $x = x'$. □

483 Using the three lemmas above, we now prove Lemma 1.

484 *Proof of Lemma 1.* By the first condition and Lemma 2, we know $f^*$ is $\frac{1}{\beta}-$strongly convex with
485 respect to $\|\cdot\|_*$. Therefore we see

$$f^*(\nabla f(y)) \ge f^*(\nabla f(x)) + \langle \nabla f^*(\nabla f(x)), \nabla f(y) - \nabla f(x) \rangle + \frac{1}{2\beta} \|\nabla f(x) - \nabla f(y)\|_*^2.$$

486 Using Lemma 3 and Lemma 4, we obtain

$$\langle y, \nabla f(y) \rangle - f(y) \ge (\langle x, \nabla f(x) \rangle - f(x)) + \langle x, \nabla f(y) - \nabla f(x) \rangle + \frac{1}{2\beta} \|\nabla f(x) - \nabla f(y)\|_*^2.$$

487 Rearranging the terms, we get

$$f(x) \ge f(y) + \langle x - y, \nabla f(y) \rangle + \frac{1}{2\beta} \|\nabla f(x) - \nabla f(y)\|_*^2,$$

488 which is the second condition.

489 The third condition follows from subtracting the second condition from the first condition. □

490 Finally, before moving the the proofs of our main results, we prove two properties of the Bregman
491 Divergence that play an important role in the analysis.

492 **Lemma 5.** $\forall a, b, c \in \mathbb{R}^d$ *and potential h, we have*
$$\langle \nabla h(b) - \nabla h(c), c - a \rangle = D_h(a||b) - D_h(a||c) - D_h(c||b).$$

493 *Proof.* By the definition of Bregman Divergence, we obtain
$$\begin{aligned}
D_h(a||b) &- D_h(a||c) - D_h(c||b) \\
&= (h(a) - h(b) - \langle \nabla h(b), a - b \rangle) - (h(a) - h(c) - \langle \nabla h(c), a - c \rangle) \\
&\quad - (h(c) - h(b) - \langle \nabla h(b), c - b \rangle) \\
&= -\langle \nabla h(b), a - b \rangle + \langle \nabla h(c), a - c \rangle + \langle \nabla h(b), c - b \rangle \\
&= (-\langle \nabla h(b), a - b \rangle + \langle \nabla h(b), c - b \rangle) + \langle \nabla h(c), a - c \rangle \\
&= \langle \nabla h(b), c - a \rangle + \langle \nabla h(c), a - c \rangle \\
&= \langle \nabla h(b) - \nabla h(c), c - a \rangle.
\end{aligned}$$

494 □

**Lemma 6.** *For all $a, b, c \in \mathbb{R}^d$, we have*

$$D_h(c||a) - D_h(c||b) = D_h(0||a) - D_h(0||b) + \langle \nabla h(b) - \nabla h(a), c \rangle.$$

*Proof.* Using the definition of Bregman divergence, we obtain

$$
\begin{aligned}
D_h(c||a) - &D_h(c||b) \\
&= h(c) - h(a) - \langle \nabla h(a), c - a \rangle - h(c) + h(b) + \langle \nabla h(b), c - b \rangle \\
&= \big(h(b) - \langle \nabla h(b), b \rangle\big) - \big(h(a) - \langle \nabla h(a), a \rangle\big) + \langle \nabla h(b) - \nabla h(a), c \rangle \\
&= D_h(0||a) - D_h(0||b) + \langle \nabla h(b) - \nabla h(a), c \rangle.
\end{aligned}
$$

$\square$

# A  Proof of Theorem 1

We consider a sequence of hitting cost functions on the real line such that the algorithm stays at the starting point through time steps $t = 1, 2, \cdots, n$ and is forced to incur a huge movement cost at time step $t = n + 1$, whereas the offline adversary can pay relatively little cost by dividing the long trek between $x_0$ and $v_{n+1}$ into multiple small steps through time steps $t = 1, 2, \cdots, n + 1$.

Specifically, suppose the starting point of the algorithm and the offline adversary is $x_0 = x_0^* = 0$, and the hitting cost functions are

$$f_t(x) = \begin{cases} \frac{m}{2} x^2 & t \in \{1, 2, \cdots, n\} \\ \frac{m'}{2}(x-1)^2 & t = n+1 \end{cases}$$

for some large parameter $m'$ that we choose later.

Suppose the algorithm first moves at time step $t_0$. If $t_0 < n + 1$, we stop the game at time step $t_0$ and compare the algorithm with an offline adversary which always stays at $x = 0$. The total cost of offline adversary is 0, but the total cost of the algorithm is non-zero. So, the competitive ratio is unbounded.

Next we consider the case where $t_0 \geq n + 1$. This implies that $x_1, \ldots x_n = 0$ and $x_{n+1}$ is some non-zero point, say $x$. We see that the cost incurred by the online algorithm is

$$cost(ALG) \geq \min_{x_{n+1}}(M_{n+1} + H_{n+1}) = \min_x \left( \frac{1}{2}x^2 + \frac{m'}{2}(x-1)^2 \right).$$

Notice that the right hand side tends to $\frac{1}{2}$ as $m'$ tends to infinity; specifically, we have

$$cost(ALG) \geq \min_x \left( \frac{1}{2}x^2 + \frac{m'}{2}(x-1)^2 \right) = \frac{1}{2\left(1 + \frac{1}{m'}\right)}. \tag{3}$$

Now let us consider the offline optimal. Notice that, in the limit as $m'$ tends to infinity, the offline optimal must satisfy $x_0^* = 0$ and $x_{n+1}^* = 1$; otherwise it would incur unbounded cost. Our lower bound is derived by considering the case when $m' \to \infty$ and so we constrain the adversary to satisfy the above, knowing that the adversary is not optimal for finite $m'$, i.e., $cost(ADV) \geq cost(OPT)$ with $cost(ADV) \to cost(OPT)$ as $m' \to \infty$.

Let the sequence of points the adversary chooses as $x^* = (x_0^*, x_1^*, \cdots, x_{n+1}^*) \in \mathbb{R}^{n+2}$. We compute the cost incurred by the adversary as follows where, to simplify presentation, we define $\mathcal{K}(n, y)$ to be the set $\{x \in \mathbb{R}^{n+2} \mid x_i \leq x_{i+1}, x_0 = 0, x_{n+1} = y\}$.

$$
\begin{aligned}
a_n &= 2 \min_{x^* \in \mathcal{K}(n,1)} \sum_{i=1}^{n+1} (H_i^* + M_i^*) \\
&= 2 \min_{x^* \in \mathcal{K}(n,1)} \left( \sum_{i=1}^{n} \frac{m}{2}(x_i^*)^2 + \sum_{i=1}^{n+1} \frac{1}{2}(x_i^* - x_{i-1}^*)^2 \right).
\end{aligned}
$$

In words, $a_n$ is twice the minimal offline cost subject to the constraints $x_0^* = 0, x_{n+1}^* = 1$. We derive the limiting behavior of the offline costs as $n \to \infty$ in the following lemma.

521  **Lemma 7.** *For $m > 0$, define*

$$a_n = 2 \min_{x^* \in \mathcal{K}(n,1)} \left( \sum_{i=1}^{n} \frac{m}{2}(x_i^*)^2 + \sum_{i=1}^{n+1} \frac{1}{2}(x_i^* - x_{i-1}^*)^2 \right).$$

522  *Then we have $\lim_{n \to \infty} a_n = \frac{-m + \sqrt{m^2 + 4m}}{2}$.*

523  Given the lemma, the total cost of the offline adversary will be $\frac{a_n}{2}$. Finally, applying (3), we know
524  $\forall n$ and $\forall m' > 0$,

$$\frac{cost(ALG)}{cost(ADV)} \geq \frac{\frac{1}{2(1+\frac{1}{m'})}}{\frac{a_n}{2}} = \frac{1}{(1+\frac{1}{m'})a_n}.$$

525  By taking the limit $n \to \infty$ and $m' \to \infty$ and using Lemma 7, we obtain

$$\frac{cost(ALG)}{cost(OPT)} = \lim_{n,m' \to \infty} \frac{cost(ALG)}{cost(ADV)} \geq \left( \frac{-m + \sqrt{m^2 + 4m}}{2} \right)^{-1} = \frac{1 + \sqrt{1 + \frac{4}{m}}}{2}.$$

526  All that remains is to prove Lemma 7, which describes the cost of the offline adversary in the limit as
527  $n$ tends to infinity.

528  *Proof of Lemma 7.* Using the fact that the costs are all homogeneous of degree 2, we see that for all
529  $y \in [0,1]$, we have

$$\min_{x^* \in \mathcal{K}(n,y)} \left( \sum_{i=1}^{n} \frac{m}{2}(x_i^*)^2 + \sum_{i=1}^{n+1} \frac{1}{2}(x_i^* - x_{i-1}^*)^2 \right)$$

$$= y^2 \min_{x^* \in \mathcal{K}(n,1)} \left( \sum_{i=1}^{n} \frac{m}{2}(x_i^*)^2 + \sum_{i=1}^{n+1} \frac{1}{2}(x_i^* - x_{i-1}^*)^2 \right). \tag{4}$$

530  The sequence $\{a_n\}, n \geq 0$ has a recursive relationship as follows:

$$a_{n+1} = 2 \min_{x^* \in \mathcal{K}(n+1,1)} \left( \sum_{i=1}^{n+1} \frac{m}{2}(x_i^*)^2 + \sum_{i=1}^{n+2} \frac{1}{2}(x_i^* - x_{i-1}^*)^2 \right)$$

$$= 2 \min_{0 \leq x \leq 1} \left( \min_{x^* \in \mathcal{K}(n,x)} \left( \sum_{i=1}^{n} \frac{m}{2}(x_i^*)^2 + \sum_{i=1}^{n+1} \frac{1}{2}(x_i^* - x_{i-1}^*)^2 \right) \right.$$

$$\left. + \frac{m}{2}x^2 + \frac{1}{2}(1-x)^2 \right)$$

$$= 2 \min_{0 \leq x \leq 1} \left( x^2 \min_{x^* \in \mathcal{K}(n,1)} \left( \sum_{i=1}^{n} \frac{m}{2}(x_i^*)^2 + \sum_{i=1}^{n+1} \frac{1}{2}(x_i^* - x_{i-1}^*)^2 \right) \right. \tag{5}$$

$$\left. + \frac{m}{2}x^2 + \frac{1}{2}(1-x)^2 \right)$$

$$= 2 \min_{0 \leq x \leq 1} \left( \frac{a_n}{2}x^2 + \frac{m}{2}x^2 + \frac{1}{2}(1-x)^2 \right)$$

$$= \frac{a_n + m}{a_n + m + 1}.$$

531  Solving the equation $x = \frac{x+m}{x+m+1}$, we find the two fixed points of the recursive relationship $a_{n+1} =$
532  $\frac{a_n+m}{a_n+m+1}$ are

$$x_1 = \frac{-m + \sqrt{m^2 + 4m}}{2},$$

and

$$x_2 = \frac{-m - \sqrt{m^2 + 4m}}{2}.$$

Notice that for $i = 1, 2$, we have

$$m - (m+1)x_i = -(1 - x_i)x_i.$$

Using this property, we obtain

$$a_{n+1} - x_1 = \frac{a_n + m}{a_n + m + 1} - x_1 = \frac{(1 - x_1)a_n + m - (m+1)x_1}{a_n + m + 1} = \frac{(1 - x_1)(a_n - x_1)}{a_n + m + 1}, \quad (6)$$

and

$$a_{n+1} - x_2 = \frac{a_n + m}{a_n + m + 1} - x_2 = \frac{(1 - x_2)a_n + m - (m+1)x_2}{a_n + m + 1} = \frac{(1 - x_2)(a_n - x_2)}{a_n + m + 1}. \quad (7)$$

Notice that $a_{n+1} - x_2 > 0$. By dividing equations (6) and (7), we obtain

$$\left( \frac{a_{n+1} - x_1}{a_{n+1} - x_2} \right) = \frac{1 - x_1}{1 - x_2} \cdot \left( \frac{a_n - x_1}{a_n - x_2} \right), \forall n \geq 0.$$

Remember that $a_0 = 1$. Therefore we have

$$\left( \frac{a_n - x_1}{a_n - x_2} \right) = \left( \frac{1 - x_1}{1 - x_2} \right)^n \left( \frac{a_0 - x_1}{a_0 - x_2} \right) = \left( \frac{1 - x_1}{1 - x_2} \right)^{n+1}.$$

Rearranging this equation, we get

$$a_n = \left( 1 - \left( \frac{1 - x_1}{1 - x_2} \right)^{n+1} \right)^{-1} \left( x_1 - x_2 \cdot \left( \frac{1 - x_1}{1 - x_2} \right)^{n+1} \right).$$

Since $0 < \left( \frac{1 - x_1}{1 - x_2} \right) < 1$, we have

$$\lim_{n \to \infty} a_n = x_1 = \frac{-m + \sqrt{m^2 + 4m}}{2}. \quad (8)$$

$\square$

# B  Proof of Theorem 2

Our proof of Theorem 2 relies on a set of technical lemmas, which follow. Lemma 8 and Lemma 10 work together to establish a lower bound on the competitive ratio as $m$ tends to zero when the balance parameter $\gamma$ is set to be $o(1/m)$, while Lemma 11 lower bound on the competitive ratio as $m$ tends to zero when the balance parameter $\gamma$ is set to be $\Omega(1/m)$.

**Lemma 8.** *If $\gamma = o(1/m)$, the competitive ratio of OBD is $\Omega(1/(\gamma m))$ when $m \to 0^+$.*

*Proof.* Our approach is to construct a scenario where OBD is forced to move along the circumference of a large circle, but the offline adversary moves along the circumference of a much smaller circle (see Figure 1). The adversary is hence able to pay much smaller movements costs, forcing the competitive ratio to be large.

We propose a series of costs which force OBD to move in a circle. The idea is to construct a cost function so that, at the end of every round, the relative positions of the OBD algorithm, the offline adversary, and the minimizer are fixed. Since OBD is memoryless, we can simply input this function arbitrarily many times and the positions of OBD and the offline adversary will trace out a pair of concentric circles (see Figure 1).

Suppose that, at the start of a round, OBD is at the point $A$. Let $\ell$ be the distance between OBD and the adversary. Consider a right triangle $ABC$ such that $|AB| = h = \sqrt{\gamma m}\ell$, the offline adversary is at some point $D$ on the hypotenuse $AC$ and $|AD| = |BC| = \ell$ (see Figure 2). Let us introduce a coordinate system such that the origin lies at $C$, the $x$-axis contains $BC$ and the $y$-axis is parallel

Figure 2: *In the right triangle $\triangle ABC$, $\angle ABC = 90^o$, $|BC| = \ell$, $|AB| = h = \sqrt{\gamma m}\ell$. Point D is on the line segment AC such that $|AD| = \ell$. OBD starts at point A and selects point E. The offline adversary starts at point D and selects point F. G is the projection point of E on line segment AB.*

to $AB$, such that the positive part of the axis lies on the same side of $BC$ as the segment $AC$. Our goal is to construct a cost function which forces OBD towards $B$. This will preserve the relative positions of OBD and the adversary, since we assumed that they were a distance $\ell$ away at the start of the round. Consider the costs $g(u) = \frac{m}{2}\|u - C\|^2$, $h(u) = \alpha \cdot d(u, BC)$ where $d(u, BC)$ is the distance from the point $u$ to the line passing through $B$ and $C$ and $\alpha > 0$ is a parameter we will pick later. Define $f_t(u) = h(u) + g(u)$. Notice that $f_t$ is $m$-strongly convex because it is the sum of an $m$-strongly convex function and a convex function. Intuitively, when $\alpha$ is large, the function $f_t$ is infinity outside of the line $BC$ but is equal to $g(u) = \frac{m}{2}\|u - C\|^2$ when restricted to points $u$ on the line. After observing the cost $f_t$, OBD will pick some new point $E$.

The following lemma highlights that $E$ can be driven arbitrarily close to $B$ by taking $\alpha$ to be sufficiently large.

**Lemma 9.** *Let $\varepsilon > 0$, and suppose $\alpha$ is picked to that $\alpha > \frac{hm\ell^2}{\varepsilon^2}$. Then the point $E$ picked by OBD satisfies $|EB| < \epsilon$.*

We instruct the adversary to pick the point $F$ on the line $BC$ (the $x$-axis) such that $EF = \ell$ (see Figure 2). Notice that $|CF| = |BF| - |BC| \le |BE| + |EF| - |BC| = |EB| + \ell - \ell < \varepsilon$, where we used the triangle inequality. Let $z = |DC|$. We see that the total cost incurred by the offline adversary is

$$M_t^* + H_t^* = \frac{1}{2}|DF|^2 + \frac{m}{2}|CF|^2 \le \frac{1}{2}(|DC| + |CF|)^2 + \frac{m}{2}|CF|^2 \le \frac{1}{2}(z + \varepsilon)^2 + \frac{m\varepsilon^2}{2},$$

where we applied the triangle inequality.

Notice that $h = |AB| = \sqrt{|AC|^2 - |BC|^2}$ by the Pythagorean theorem (recall that $ABC$ is a right triangle). Since $|AC| = \ell + z$ and $|BC| = \ell$, we see that $h = \sqrt{2z\ell + z^2}$. Hence the movement cost incurred by the OBD is

$$M_t \ge \frac{1}{2}(h - \varepsilon)^2 = \frac{1}{2}(\sqrt{2z\ell + z^2} - \varepsilon)^2.$$

Hence the ratio of the costs is

$$\frac{M_t + H_t}{M_t^* + H_t^*} \ge \frac{M_t}{M_t^* + H_t^*} \ge \frac{\frac{1}{2}(\sqrt{2z\ell + z^2} - \varepsilon)^2}{\frac{1}{2}(z + \varepsilon)^2 + \frac{m\varepsilon^2}{2}}.$$

Since the limit of this expression as $\varepsilon \to 0$ is $\frac{2z\ell + z^2}{z^2}$, for sufficiently small $\varepsilon$ this will be at least $\frac{1}{2}\frac{2z\ell + z^2}{z^2} \ge \frac{\ell}{z}$. Since $z = \sqrt{h^2 + \ell^2} - \ell$ and $h = \sqrt{\gamma m}\ell$, the ratio of costs is at least

$$\frac{\ell}{\sqrt{\gamma m\ell^2 + \ell^2} - \ell} = \frac{1}{\sqrt{\gamma m + 1} - 1} = \frac{\sqrt{\gamma m + 1} + 1}{\gamma m} \ge \frac{2}{\gamma m}.$$

Now, we describe the whole process. When $t = 1$, the hitting cost function is $f_1(x) = \frac{m}{2}\|x\|_2^2$. While OBD stays at $x = 0$, the adversary moves to the point $(\ell, 0)$; it incurs a one-time cost of

$$x_{t-1} \qquad x_t \qquad v_t = t$$

Figure 3: *Balance condition at time step $t$ in Lemma 10. Starting from $x_{t-1}$, OBD picks $x_t$ after observing the hitting cost function $f_t(x) = \frac{m}{2}(x-t)^2$.*

$M_1^* + H_1^* = \frac{1}{2}\ell^2 + \frac{m}{2}\ell^2$. On all subsequent steps $t = 2 \ldots T$, we repeatedly apply the construction, which forces OBD to move in a circle. The one-time cost incurred by the adversary to setup the game is negligible in the limit as $T$ is large, and the per-round ratio of costs is $\Omega(\frac{1}{\gamma m})$, so the competitive ratio is also $\Omega(\frac{1}{\gamma m})$ as claimed. $\qquad\square$

The key technical lemma used in the proof is Lemma 9, and we now provide a proof of that result.

*Proof of Lemma 9.* Suppose $\alpha > \frac{hm\ell^2}{\varepsilon^2}$. We first show that OBD selects the point $E$ strictly contained by the $\frac{m}{2}\ell^2$-level set, which is the one $B$ lies on. First observe that the point $B$ satisfies the balance condition: $\frac{1}{2}|AB|^2 = \gamma\frac{m}{2}|BC|^2$, because we constructed $ABC$ so that $|AB| = h = \sqrt{\gamma m}\ell$ and $|BC| = \ell$. However, the point $B$ is not necessarily a projection of $A$ onto any level set of $f_t$. If OBD projected onto the level set which $B$ lies on, it would incur less cost than if it moved to $B$; however then the balance condition would be violated. To restore the balance condition, we must increase the movement cost while decreasing the hitting cost – which means we must move to a strictly smaller level set, say the $\frac{m}{2}l_1^2$-level set, where $l_1 < l$.

Let $E_y$ denote the $y$-coordinate of $E$, using the coordinate system we define in the proof of Lemma 8. Notice that $E_y = \frac{g(E)}{\alpha}$, since $g(E)$ was defined to be the vertical distance to the $x$-axis times $\alpha$. Since $g(E) \leq f_t(E)$, we see that $E_y \leq \frac{f_t(E)}{\alpha} = \frac{ml_1^2}{2\alpha} \leq \frac{ml^2}{2\alpha}$, where we used the fact that $E$ lies on the $\frac{m}{2}\ell_1^2$ level set and $\ell_1 \leq \ell$. By the balance condition, $\frac{1}{2}|AE|^2 = \frac{\gamma m}{2}l_1^2 \leq \frac{\gamma m}{2}l^2 = \frac{1}{2}h^2$. Let $G$ be the point with coordinates $(B_x, E_y)$. Applying the Pythagorean theorem successively to the right triangle $BEG$ and the right triangle $AEG$, we see that

$$
\begin{aligned}
|EB|^2 = |E_x - B_x|^2 + E_y^2 &\leq (|AE|^2 - (|AB| - E_y)^2) + E_y^2 \\
&\leq (|AB|^2 - (|AB| - E_y)^2) + E_y^2 \leq 2h \cdot E_y \leq h\frac{ml^2}{\alpha},
\end{aligned}
\tag{9}
$$

where we used the fact that $|AB| \geq |AE|$ and $|AB| = h$. Since we picked $\alpha > \frac{hm\ell^2}{\varepsilon^2}$, we see that $|EB| < \varepsilon$.

$\qquad\square$

Now we move on to the next technical lemma in the proof of Theorem 2.

**Lemma 10.** *When $\gamma = o(\frac{1}{m})$, the competitive ratio of OBD is $\Omega(\sqrt{\frac{\gamma}{m}})$.*

*Proof.* We consider a sequence of cost functions on the real line such that the OBD algorithm moves far away from the starting point, incurring significant movement costs, whereas the offline adversary could pay relatively little cost by staying at the starting point. More specifically, we consider the sequence of hitting cost functions $f_t(x) = \frac{m}{2}(x-t)^2, t = 1, 2, \cdots, n$. The value of $n$ will be picked later. We assume the starting point is at zero.

Notice that by the balance condition we always have $M_t = \gamma H_t$, so $\frac{1}{2}\|x_t - x_{t-1}\|^2 = \gamma\frac{m}{2}\|x_t - t\|^2$. We can rearrange this expression to obtain $\frac{x_t - x_{t-1}}{t - x_t} = \sqrt{\gamma m}$. Define

$$\lambda = \frac{x_t - x_{t-1}}{t - x_{t-1}} = \frac{\sqrt{\gamma m}}{1 + \sqrt{\gamma m}}.$$

We obtain the recursive equation $x_t = x_{t-1} + (t - x_{t-1})\lambda$ with initial condition $x_0 = 0$. Solving this equation, we obtain $x_t = t - \frac{1-\lambda}{\lambda}(1 - (1-\lambda)^t)$.

Suppose we picked $n$ to be $= \lceil \frac{1}{\lambda} \rceil$. By assumption, $\gamma = o(\frac{1}{m})$; hence in the limit as $m$ tends to zero, $\lambda$ also tends to zero. Notice that $x_n = n - \frac{1-\lambda}{\lambda}(1 - (1-\lambda)^n) \geq \frac{1}{\lambda}\frac{1}{2e} - (1 - \frac{1}{e}) \geq \frac{1}{6\lambda}$ for sufficiently small $\lambda$. Here we used the fact that $(1-\lambda)^{\frac{1}{\lambda}} \to e^{-1}$.

Suppose the next cost function is $f_{n+1}(x) = m'x^2$. Notice that if the offline adversary simply stays at zero throughout the game, the total cost it incurs would be

$$cost(ADV) = \frac{m}{2}(1^2 + 2^2 + \cdots + n^2) \leq \frac{mn^3}{2} = \Theta\left(\frac{m}{\lambda^3}\right) = \Theta\left(\frac{1}{\sqrt{\gamma^3 m}}\right).$$

In the last step, we used the fact that $\lambda$ tends to $\sqrt{\gamma m}$ when $\gamma = o(\frac{1}{m})$ and $m$ tends to zero.

If we pick $m'$ large enough that OBD is forced to incur movement cost at least $\frac{1}{2}(\frac{x_n}{2})^2$, the total cost incurred by OBD is

$$cost(OBD) \geq \frac{1}{2}\left(\frac{x_n}{2}\right)^2 = \Theta\left(\frac{1}{\lambda^2}\right) = \Theta\left(\frac{1}{\gamma m}\right).$$

Putting these facts together, we see that the competitive ratio is at least $\Theta(\sqrt{\frac{\gamma}{m}})$. $\qquad\square$

The last technical lemma used to proof Theorem 2 is the following.

**Lemma 11.** *When $\gamma = \Omega(\frac{1}{m})$, the competitive ratio of OBD is $\Omega\left(\frac{1}{m}\right)$.*

*Proof.* Since $\gamma = \Omega(\frac{1}{m})$, we can assume there exists $C > 0$ such that $\gamma \geq C/m$. We again consider a situation such that the OBD algorithm moves far away from the starting point, incurring significant movement cost, whereas the offline adversary could pay relatively little cost by staying at the starting point. More specifically, suppose the starting point is zero and the first cost function is $f_1(x) = \frac{m}{2}(1-x)^2$. Suppose the adversary stays at zero. The cost incurred by the adversary will be

$$cost(ADV) = \frac{m}{2}.$$

Notice that by the balance condition ($M_t = \gamma H_t$), the point $x_1$ picked by OBD satisfies $\frac{x_1^2}{2} = \gamma \frac{m}{2}(1-x_1)^2$. So the cost incurred by OBD is lower bounded by

$$cost(OBD) \geq M_1 = \frac{1}{2}\left(\frac{\sqrt{\gamma m}}{1 + \sqrt{\gamma m}}\right)^2 \geq \frac{1}{2}\left(\frac{\sqrt{C}}{1 + \sqrt{C}}\right)^2.$$

Since $C$ is a positive constant, the competitive ratio of OBD is lower bounded by $\frac{OBD}{ADV} = \Theta\left(\frac{1}{m}\right)$. $\quad\square$

Now we return to the proof of Theorem 2. This proof is a straightforward combination of the above lemmas. When $\gamma = o(\frac{1}{m})$, by combining Lemma 8 and Lemma 10, we know the competitive ratio is at least $\max\left(\frac{C_1}{\gamma m}, C_2\sqrt{\frac{\gamma}{m}}\right)$ for some positive constants $C_1, C_2$. Notice that function $\frac{C_1}{\gamma m}$ is monotonically decreasing in $\gamma$ and $C_2\sqrt{\frac{\gamma}{m}}$ is monotonically increasing in $\gamma$. Solving the equation $\frac{C_1}{\gamma m} = C_2\sqrt{\frac{\gamma}{m}}$, we get $\gamma = \left(\frac{C_1}{C_2}\right)^{\frac{2}{3}} m^{-\frac{1}{3}}$. Therefore we see that

$$\max\{\frac{C_1}{\gamma m}, C_2\sqrt{\frac{\gamma}{m}}\} \geq C_1^{\frac{1}{3}} C_2^{\frac{2}{3}} m^{-\frac{2}{3}} = \Theta(m^{-\frac{2}{3}}).$$

On the other hand, when $\gamma = \Omega(\frac{1}{m})$, by Lemma 11, we know the competitive ratio of OBD is lower bounded by $\Theta\left(\frac{1}{m}\right)$.

Together, the above implies that the competitive ratio of OBD is at least $\Theta(m^{-\frac{2}{3}})$ when $m \to 0^+$.

 # C  Proof of Theorem 3

To begin, note that it is sufficient to prove result for all positive $m \leq \frac{9}{64}$. Similarly, it also suffices to show Theorem 3 when the minimum of every hitting cost function is zero, since otherwise the competitive ratio can only improve if this is not the case.

Our argument makes use of the following potential function: $\phi(x_t, x_t^*) = \eta \left\| x_t - x_t^* \right\|^2$. We define $\Delta\phi = \phi(x_t, x_t^*) - \phi(x_{t-1}, x_{t-1}^*)$ and $\Delta\phi' = \phi(x_t', x_t^*) - \phi(x_{t-1}, x_{t-1}^*)$. It suffices to show that $H_t + M_t + \Delta\phi \leq C(H_t^* + M_t^*)$, for some positive constant $C$. From this inequality, we can sum over all timesteps $t$ to yield that the competitive ratio is upper bounded by $C$:

$$\sum_{t=0}^{T} H_t + M_t \leq \sum_{t=0}^{T} H_t + M_t + \Delta\phi \leq C \sum_{t=0}^{T} (H_t^* + M_t^*).$$

Throughout the proof, we fix $\eta = 4$ and use $\|\cdot\|$ to denote $\ell_2$ norm. When we refer to generalized mean inequality, we mean

$$(a + b)^2 \leq 2a^2 + 2b^2, \forall a, b \in \mathbb{R}.$$

We define $H_t' := f_t(x_t')$ and $M_t' := c(x_t', x_{t-1}) = \frac{1}{2} \left\| x_t' - x_{t-1} \right\|_2^2$, where $x_t'$ is the point chosen by the first OBD phase (line 3) of Algorithm 2.

Before we move to the main casework in the proof, we begin with a technical lemma that we use to bound the change in the potential function.

**Lemma 12.** *Suppose the potential function $\phi : \mathbb{R}^d \times \mathbb{R}^d \to \mathbb{R}_{\geq 0}$ is defined as $\phi(a, b) = \eta \left\| a - b \right\|^2$, where $\eta > 0$. Then $\forall \lambda > 0$, the change in potential satisfies*

$$\phi(a, c) - \phi(a, b) \leq (1 + \lambda^2)\phi(b, c) + \frac{1}{\lambda^2}\phi(a, b),$$

*for all $a, b, c \in \mathbb{R}^d$.*

*Proof.* Using the triangle inequality, we obtain

$$\left\| a - c \right\|^2 \leq (\left\| a - b \right\| + \left\| b - c \right\|)^2 = \left\| a - b \right\|^2 + \left\| b - c \right\|^2 + 2 \left\| a - b \right\| \left\| b - c \right\|.$$

Rearranging the terms, we obtain

$$\begin{aligned}
\left\| a - c \right\|^2 - \left\| a - b \right\|^2 &\leq \left\| b - c \right\|^2 + 2 \left\| a - b \right\| \left\| b - c \right\| \\
&= \left\| b - c \right\|^2 + 2(\frac{1}{\lambda} \left\| a - b \right\|)(\lambda \left\| b - c \right\|) \\
&\leq (1 + \lambda^2) \left\| b - c \right\|^2 + \frac{1}{\lambda^2} \left\| a - b \right\|^2,
\end{aligned}$$

where in the last line we use the AM-GM inequality. $\qquad\square$

We are now ready to precede with the proof, which is divided up into two cases based on the relationship between the hitting cost of the algorithm and that of the adversary.

**Case 1:** $H_t' \leq H_t^*$

Since the hitting cost function satisfies $f_t(x) \geq \frac{m}{2} \left\| x - v_t \right\|^2$, by the triangle inequality, we have

$$\left\| x_t' - x_t^* \right\| \leq \left\| x_t' - v_t \right\| + \left\| x_t^* - v_t \right\| \leq \left( \sqrt{\frac{2H_t'}{m}} + \sqrt{\frac{2H_t^*}{m}} \right). \tag{10}$$

 Thus the change in potential satisfies

$$
\frac{1}{\eta}\Delta\phi' = \left\|x_t' - x_t^*\right\|^2 - \left\|x_{t-1} - x_{t-1}^*\right\|^2
$$

$$
= (\left\|x_t' - x_t^*\right\| - \left\|x_{t-1} - x_{t-1}^*\right\|)(\left\|x_t' - x_t^*\right\| + \left\|x_{t-1} - x_{t-1}^*\right\|)
$$

$$
\leq (\left\|x_t' - x_{t-1}\right\| + \left\|x_t^* - x_{t-1}^*\right\|)\left(\left\|x_t' - x_{t-1}\right\| + \left\|x_t^* - x_{t-1}^*\right\| + 2\left\|x_t' - x_t^*\right\|\right) \quad \text{(11a)}
$$

$$
= (\left\|x_t' - x_{t-1}\right\| + \left\|x_t^* - x_{t-1}^*\right\|)^2 + 2(\left\|x_t' - x_{t-1}\right\| + \left\|x_t^* - x_{t-1}^*\right\|)\left\|x_t' - x_t^*\right\|
$$

$$
\leq 2\left\|x_t' - x_{t-1}\right\|^2 + 2\left\|x_t^* - x_{t-1}^*\right\|^2 + 2(\left\|x_t' - x_{t-1}\right\| + \left\|x_t^* - x_{t-1}^*\right\|)\left\|x_t' - x_t^*\right\| \quad \text{(11b)}
$$

$$
\leq 4M_t' + 4M_t^* + 2(\sqrt{2M_t'} + \sqrt{2M_t^*})\left(\sqrt{\frac{2H_t'}{m}} + \sqrt{\frac{2H_t^*}{m}}\right) \quad \text{(11c)}
$$

$$
\leq 4M_t' + 4M_t^* + \sqrt{\frac{1}{m}}((\sqrt{2M_t'} + \sqrt{2M_t^*})^2 + (\sqrt{2H_t'} + \sqrt{2H_t^*})^2) \quad \text{(11d)}
$$

$$
\leq 4M_t' + 4M_t^* + \sqrt{\frac{1}{m}}((4M_t' + 4M_t^*) + (4H_t' + 4H_t^*)) \quad \text{(11e)}
$$

$$
= \left(4 + 4\sqrt{\frac{1}{m}}\right)M_t' + \left(4 + 4\sqrt{\frac{1}{m}}\right)M_t^* + 4\sqrt{\frac{1}{m}}H_t' + 4\sqrt{\frac{1}{m}}H_t^*,
$$

 where we use the triangle inequality in line (11a); the generalized mean inequality in lines (11b),
 (11d) and (11e) and inequality (10) in line (11c).

 Using the OBD's balance condition $M_t' = \gamma H_t'$ and the assumption $H_t' \leq H_t^*$ based on inequality
 (11), we have

$$
\frac{1}{\eta}\Delta\phi' \leq \left(4 + 4\sqrt{\frac{1}{m}}\right)\gamma H_t' + \left(4 + 4\sqrt{\frac{1}{m}}\right)M_t^* + 4\sqrt{\frac{1}{m}}H_t' + 4\sqrt{\frac{1}{m}}H_t^*
$$

$$
\leq \left(4 + 4\sqrt{\frac{1}{m}}\right)\gamma H_t^* + \left(4 + 4\sqrt{\frac{1}{m}}\right)M_t^* + 8\sqrt{\frac{1}{m}}H_t^*.
$$

 Notice that by the triangle inequality and the generalized mean inequality, we have that

$$
M_t = \frac{1}{2}\left\|x_t - x_{t-1}\right\|^2 \leq \frac{1}{2}(\left\|x_t' - x_{t-1}\right\| + \left\|x_t - x_t'\right\|)^2 \leq \frac{1}{2}(2\left\|x_t' - x_{t-1}\right\|^2 + 2\left\|x_t - x_t'\right\|^2).
$$

Remember that since $\mu = 1$, we have $\|x_t - x_t'\|^2 = m\|x_t' - v_t\|^2$. Using this fact, we derive the following bound on $H_t + M_t + \Delta\phi$:

$$
\begin{aligned}
H_t + M_t + \Delta\phi &\leq H_t' + \frac{1}{2}\left(2\|x_t' - x_{t-1}\|^2 + 2\|x_t - x_t'\|^2\right) \\
&\quad + \eta(\|x_t - x_t^*\|^2 - \|x_t' - x_t^*\|^2) + \Delta\phi' \\
&\leq H_t' + \left(2M_t' + m\|x_t' - v_t\|^2\right) \\
&\quad + \left(\eta\left(1 + \frac{1}{\sqrt{m}}\right)\|x_t - x_t'\|^2 + \eta\sqrt{m}\|x_t' - x_t^*\|^2\right) + \Delta\phi' \quad\quad (12\mathrm{a}) \\
&\leq H_t' + \left(2M_t' + m\|x_t' - v_t\|^2\right) \\
&\quad + \left(\eta\left(1 + \frac{1}{\sqrt{m}}\right)m\|x_t' - v_t\|^2 + \eta\sqrt{m}\left(2\|x_t' - v_t\|^2 + 2\|x_t^* - v_t\|^2\right)\right) \\
&\quad + \Delta\phi' \quad\quad (12\mathrm{b}) \\
&\leq H_t' + (2M_t' + 2H_t') + \left(\eta\left(1 + \frac{1}{\sqrt{m}}\right)2H_t' + \eta\sqrt{m}\left(\frac{4H_t'}{m} + \frac{4H_t^*}{m}\right)\right) + \Delta\phi' \\
&\quad\quad (12\mathrm{c}) \\
&= (3 + 2\eta + \frac{6\eta}{\sqrt{m}})H_t' + 2M_t' + 4\eta\frac{H_t^*}{\sqrt{m}} + \Delta\phi' \\
&= \left(3 + 2\eta + \frac{6\eta}{\sqrt{m}} + 2\gamma\right)H_t' + 4\eta\frac{H_t^*}{\sqrt{m}} + \Delta\phi' \\
&\leq \left(3 + 2\eta + \frac{6\eta}{\sqrt{m}} + 2\gamma\right)H_t^* + 4\eta\frac{H_t^*}{\sqrt{m}} + \Delta\phi' \quad\quad (12\mathrm{d}) \\
&= \left(3 + 2\eta + \frac{10\eta}{\sqrt{m}} + 2\gamma\right)H_t^* + \Delta\phi',
\end{aligned}
$$

where we use Lemma 12 in line (12a); the triangle inequality in line (12b); $m$-strongly convexity of $f_t$ in line (12c); and the assumption $H_t' \leq H_t^*$ in line (12d).

Combining inequalities (11) and (12), we obtain

$$
H_t + M_t + \Delta\phi \leq \left(3 + 2\eta + 2\gamma + 4\eta\gamma + \frac{\eta}{\sqrt{m}}(18 + 4\gamma)\right)H_t^* + \eta(4 + 4\sqrt{\frac{1}{m}})M_t^*. \quad\quad (13)
$$

**Case 2:** $H_t' \geq H_t^*$

In this case, we prove that for any $x_t^*, x_{t-1}^* \in \mathbb{R}^d$, we have

$$
H_t + M_t + \Delta\phi \leq \frac{C}{\sqrt{m}}(H_t^* + M_t^*), \quad\quad (14)
$$

for some positive constant $C$.

In the proof, we use $D_1, D_2, \cdots, D_d$ to represent the $d$ axes in the coordinate system.

As shown in Figure 4, without loss of generality, let $v_t = (0, 0, \cdots, 0), x_t' = (h_1, h_2, 0, \cdots, 0)$ and $D_2 = h_2$ be the projection hyper plane, where $h_1 \geq 0, h_2 \geq 0$. And let $l = \|x_{t-1} - x_t'\| > 0$. Note that our analysis still holds in one-dimension because we can restrict ourselves to the $D_2$ axis.

Then we know $x_{t-1} = (h_1, h_2 + l, 0, \cdots, 0), x_t = (h_1(1 - \sqrt{m}), h_2(1 - \sqrt{m}), 0, \cdots, 0)$. Since we know $x_t^*$ must lie below the projection hyper plane, we can let $x_t^* = (x, h_2 - y, a_3, a_4, \cdots, a_d)$, where $y > 0$.

Now we show that it suffices to prove the statement when $x_{t-1}^*$ is on the line segment $x_t^* x_{t-1}$. Suppose $x_{t-1}^*$ is not on the line segment $x_t^* x_{t-1}$. If $\|x_{t-1}^* - x_{t-1}\| > \|x_t^* - x_{t-1}\|$, by moving $x_{t-1}^*$ to $x_t^*$, $\Delta\phi$ increases and $M_t^*$ decreases. Otherwise, we can choose a point $K$ on line segment $x_t^* x_{t-1}$ such that $\|K - x_{t-1}\| = \|x_{t-1}^* - x_{t-1}\|$. By moving $x_{t-1}^*$ to $K$, $\Delta\phi$ remains unchanged and $M_t^*$ decreases. Therefore if inequality (14) holds for $x_{t-1}^*$ on the segment $x_t^* x_{t_1}$, then it must also hold for any other $x_{t-1}^* \in \mathbb{R}^d$.

Figure 4: *Starting at $x_{t-1}$, G-OBD first does projection on to the $H'_t$ level set (red dashed line) in the first phase. The projection point is $x'_t$. Then G-OBD moves toward the minimizer to obtain point $x_t$ in the second phase. Let the minimizer $v_t$ be the origin. Notice that the three points $x_{t-1}, x'_t, v_t$ defines a plane $S$. Without loss of generality, we can let axis $D_2$ be parallel to line $x'_t x_{t-1}$; and let axis $D_1$ be parallel to the projection hyperplane.*

Now we suppose $x^*_{t-1}$ is on the line segment $x^*_t x_{t-1}$, and $\left\| x^*_t - x^*_{t-1} \right\| = \lambda \left\| x^*_t - x_{t-1} \right\|$.

Recall that we set $\gamma = 1$, so $M'_t = \gamma H'_t = H'_t$. It follows that

$$M_t \le l^2 + \left\| x_t - x'_t \right\|^2 = l^2 + m(h_1^2 + h_2^2) \le l^2 + 2H'_t = l^2 + 2M'_t \le 2l^2,$$

and

$$H_t \le H'_t = M'_t = \frac{l^2}{2}.$$

We can separate $\Delta \phi$ into two parts:

$$\frac{\Delta \phi}{\eta} = \left( \left\| x^*_t - x_t \right\|^2 - \left\| x^*_t - x_{t-1} \right\|^2 \right) + \left( \left\| x^*_t - x_{t-1} \right\|^2 - \left\| x^*_{t-1} - x_{t-1} \right\|^2 \right).$$

For convenience, we define

$$\Delta \phi_1 := \left( \left\| x^*_t - x_t \right\|^2 - \left\| x^*_t - x_{t-1} \right\|^2 \right),$$

and

$$\Delta \phi_2 := \left( \left\| x^*_t - x_{t-1} \right\|^2 - \left\| x^*_{t-1} - x_{t-1} \right\|^2 \right).$$

We further notice that from the triangle inequality,

$$\Delta \phi_2 \le (1 - (1 - \lambda)^2) \left\| x^*_t - x_{t-1} \right\|^2 = \lambda(2 - \lambda) \left( (x - h_1)^2 + (y + l)^2 + \sum_{i=3}^d a_i^2 \right). \tag{15}$$

Now we express $M^*_t$ and $H^*_t$ in terms of the variables we define, which are

$$M^*_t = \frac{1}{2} (\lambda \left\| x^*_t - x_{t-1} \right\|)^2 = \frac{\lambda^2}{2} \left( (x - h_1)^2 + (y + l)^2 + \sum_{i=3}^d a_i^2 \right), \tag{16}$$

and

$$H^*_t \ge \frac{m}{2} \left\| x^*_t - v_t \right\|^2 = \frac{m}{2} \left( x^2 + (h_2 - y)^2 + \sum_{i=3}^d a_i^2 \right). \tag{17}$$

We also expand $\Delta \phi_1$:

$$
\begin{aligned}
\Delta \phi_1 &= \left\| x^*_t - x_t \right\|^2 - \left\| x^*_t - x_{t-1} \right\|^2 \\
&= (x - h_1 + h_1 \sqrt{m})^2 + (y - h_2 \sqrt{m})^2 + \sum_{i=3}^d a_i^2 - (x - h_1)^2 - (y + l)^2 - \sum_{i=3}^d a_i^2 \\
&= \left( (x - h_1 + h_1 \sqrt{m})^2 - (x - h_1)^2 \right) + \left( (y - h_2 \sqrt{m})^2 - (y + l)^2 \right) \\
&= h_1 \sqrt{m}(2x - 2h_1 + h_1 \sqrt{m}) - (h_2 \sqrt{m} + l)(2y + l - h_2 \sqrt{m}) \\
&= 2x h_1 \sqrt{m} - 2h_1^2 \sqrt{m} + h_1^2 m - 2y(h_2 \sqrt{m} + l) - l^2 + h_2^2 m.
\end{aligned}
\tag{18}
$$

Using the condition that $m \le \frac{9}{64} < 1$, we derive the following bound:

$$\Delta\phi_1 \le 2xh_1\sqrt{m} - 2h_1^2\sqrt{m} + h_1^2 m - 2y(h_2\sqrt{m} + l) - l^2 + h_2^2\sqrt{m}$$
$$= 2xh_1\sqrt{m} - 2h_1^2\sqrt{m} + h_1^2 m + \sqrt{m}(h_2 - y)^2 - \sqrt{m}y^2 - 2yl - l^2. \tag{19}$$

Substituting equations (16) and (17) into inequality (14), we know that it suffices to show that for some constant $C$,

$$M_t + H_t + \eta\Delta\phi_1 + \eta\Delta\phi_2 \le \frac{C}{\sqrt{m}}\left(\frac{m}{2}\Big(x^2 + (h_2 - y)^2 + \sum_{i=3}^{d} a_i^2\Big) + \frac{\lambda^2}{2}\Big((x-h_1)^2 + (y+l)^2 + \sum_{i=3}^{d} a_i^2\Big)\right). \tag{20}$$

**Subcase 2.1:** $\lambda \le \frac{\sqrt{m}}{2}$

We can bound equation (15) as follows:

$$\Delta\phi_2 = \lambda(2-\lambda)\left((x-h_1)^2 + (y+l)^2 + \sum_{i=3}^{d} a_i^2\right)$$
$$\le \sqrt{m}(x-h_1)^2 + \sqrt{m}(y+l)^2 + \sqrt{m}\sum_{i=3}^{d} a_i^2 \tag{21}$$
$$= \sqrt{m}x^2 - 2\sqrt{m}xh_1 + \sqrt{m}h_1^2 + \sqrt{m}y^2 + 2\sqrt{m}yl + \sqrt{m}l^2 + \sqrt{m}\sum_{i=3}^{d} a_i^2.$$

Summing inequalities (19) and (21), we get

$$\Delta\phi_1 + \Delta\phi_2 \le \sqrt{m}x^2 + (-h_1^2\sqrt{m} + h_1^2 m) + \sqrt{m}(h_2 - y)^2$$
$$+ (2\sqrt{m}yl - 2yl) + (\sqrt{m}l^2 - l^2) + \sqrt{m}\sum_{i=3}^{d} a_i^2$$
$$\le \sqrt{m}x^2 + 0 + \sqrt{m}(h_2-y)^2 + 0 - \frac{5}{8}l^2 + \sqrt{m}\sum_{i=3}^{d} a_i^2 \tag{22a}$$
$$\le \sqrt{m}x^2 + \sqrt{m}(h_2-y)^2 - \frac{5}{8}l^2 + \sqrt{m}\sum_{i=3}^{d} a_i^2,$$

where we use the condition that $m \le \frac{9}{64}$ in line (22a). We further obtain

$$M_t + H_t + \eta(\Delta\phi_1 + \Delta\phi_2) \le 2l^2 + \frac{l^2}{2} + \eta\left(\sqrt{m}x^2 + \sqrt{m}(h_2-y)^2 - \frac{5}{8}l^2 + \sqrt{m}\sum_{i=3}^{d} a_i^2\right)$$
$$= \frac{5l^2}{2} + 4\left(\sqrt{m}x^2 + \sqrt{m}(h_2-y)^2 - \frac{5}{8}l^2 + \sqrt{m}\sum_{i=3}^{d} a_i^2\right)$$
$$= 4\left(\sqrt{m}x^2 + \sqrt{m}(h_2-y)^2 + \sqrt{m}\sum_{i=3}^{d} a_i^2\right).$$

Therefore, for $C \ge 8$, we have

$$M_t + H_t + \eta\Delta\phi_1 + \eta\Delta\phi_2 \le \frac{C}{\sqrt{m}}\left(\frac{m}{2}\Big(x^2 + (h_2-y)^2 + \sum_{i=3}^{d} a_i^2\Big) + \frac{\lambda^2}{2}\Big((x-h_1)^2 + (y+l)^2 + \sum_{i=3}^{d} a_i^2\Big)\right),$$

which establishes inequality (20).

**Subcase 2.2:** $\lambda \geq \frac{\sqrt{m}}{2}$

Notice that when $C \geq 32$, we have

$$\frac{C}{2\sqrt{m}}\lambda^2 \geq \frac{16}{\sqrt{m}}\lambda^2 \geq \frac{16}{\sqrt{m}} \cdot \frac{\sqrt{m}}{2}\lambda = 8\lambda \geq 4\lambda(2-\lambda) = \eta\lambda(2-\lambda).$$

Substituting this inequality into equation (15), we know that for $C \geq 32$,

$$\eta\Delta\phi_2 \leq \frac{C}{\sqrt{m}} \cdot \frac{\lambda^2}{2}\left((x-h_1)^2 + (y+l)^2 + \sum_{i=3}^{d}a_i^2\right). \tag{23}$$

We can further bound inequality (19):

$$\begin{aligned}
\Delta\phi_1 &\leq 2xh_1\sqrt{m} - 2h_1^2\sqrt{m} + h_1^2m + \sqrt{m}(h_2-y)^2 - \sqrt{m}y^2 - 2yl - l^2 \\
&\leq \sqrt{m}x^2 + \sqrt{m}h_1^2 - 2h_1^2\sqrt{m} + h_1^2m + \sqrt{m}(h_2-y)^2 - l^2 \\
&\leq \sqrt{m}x^2 + \sqrt{m}(h_2-y)^2 - l^2,
\end{aligned}$$

where we apply the AM-GM inequality in step 2 and use the condition $m < 1$ in step 3.

Therefore we have

$$\begin{aligned}
H_t + M_t + \eta\Delta\phi_1 &\leq \frac{5l^2}{2} + 4(\sqrt{m}x^2 + \sqrt{m}(h_2-y)^2 - l^2) \\
&\leq 4(\sqrt{m}x^2 + \sqrt{m}(h_2-y)^2).
\end{aligned} \tag{24}$$

Summing inequalities (24) and (23), we yield that for $C \geq 32$,

$$M_t + H_t + \eta\Delta\phi_1 + \eta\Delta\phi_2 \leq \frac{C}{\sqrt{m}}\left(\frac{m}{2}\left(x^2 + (h_2-y)^2 + \sum_{i=3}^{d}a_i^2\right) + \frac{\lambda^2}{2}\left((x-h_1)^2 + (y+l)^2 + \sum_{i=3}^{d}a_i^2\right)\right),$$

which establishes inequality (20).

Combining all cases above, we conclude that G-OBD is an $O(\frac{1}{\sqrt{m}})$-competitive algorithm.

## D  Proof of Theorem 4

To prove Theorem 4 we make use of Lemma 1 and 5.

Our approach is to make use of strong convexity and properties of Bregman Divergences to derive an inequality in the form of $H_t + M_t + \Delta\phi \leq C(H_t^* + M_t^*)$ for some positive constant $C$, where $\Delta\phi$ is the change in potential, which we will define later. The constant $C$ is then an upper bound for the competitive ratio.

To begin, recall that $h$ is assumed to be $\alpha$-strongly convex and $\beta$-strongly smooth with respect to norm $\|\cdot\|$. Thus we can give a trivial bound on Bregman Divergence, namely

$$\forall x, y, \frac{\alpha}{2}\|x-y\|^2 \leq D_h(x\|y) \leq \frac{\beta}{2}\|x-y\|^2. \tag{25}$$

Recall that the update rule in Algorithm 3 can be stated as:

$$x_t = \arg\min_x f_t(x) + \lambda_1 D_h(x\|x_{t-1}) + \lambda_2 D_h(x\|v_t).$$

Since the function $f_t(x) + \lambda_1 D_h(x\|x_{t-1}) + \lambda_2 D_h(x\|v_t)$ is strongly convex, the minimizer $x_t$ exists and is unique. Furthermore, it must satisfy the first-order condition

$$\nabla f_t(x_t) + \lambda_1(\nabla h(x_t) - \nabla h(x_{t-1})) + \lambda_2(\nabla h(x_t) - \nabla h(v_t)) = 0.$$

Further, since $f_t(x)$ is $m$-strongly convex, we have

$$\begin{aligned}
f_t(x_t^*) &\geq f_t(x_t) + \langle \nabla f_t(x_t), x_t^* - x_t \rangle + \frac{m}{2}\|x_t^* - x_t\|^2 \\
&= f_t(x_t) - \lambda_1\langle \nabla h(x_{t-1}) - \nabla h(x_t), x_t - x_t^* \rangle \\
&\quad - \lambda_2\langle \nabla h(v_t) - \nabla h(x_t), x_t - x_t^* \rangle + \frac{m}{2}\|x_t^* - x_t\|^2.
\end{aligned} \tag{26}$$

Using Lemma 5, we obtain

$$\langle \nabla h(x_{t-1}) - \nabla h(x_t), x_t - x_t^* \rangle = D_h(x_t^* || x_{t-1}) - D_h(x_t^* || x_t) - D_h(x_t || x_{t-1}),$$

and

$$\langle \nabla h(v_t) - \nabla h(x_t), x_t - x_t^* \rangle = D_h(x_t^* || v_t) - D_h(x_t^* || x_t) - D_h(x_t || v_t).$$

Substituting the two above identities into inequality (26), we get

$$f_t(x_t) + \lambda_1 D_h(x_t || x_{t-1}) + \lambda_2 D_h(x_t || v_t) + (\lambda_1 + \lambda_2) D_h(x_t^* || x_t) + \frac{m}{2} \|x_t^* - x_t\|^2$$
$$\leq f_t(x_t^*) + \lambda_1 D_h(x_t^* || x_{t-1}) + \lambda_2 D_h(x_t^* || v_t).$$

It follows that

$$f_t(x_t) + \lambda_1 D_h(x_t || x_{t-1}) + (\lambda_1 + \lambda_2) D_h(x_t^* || x_t) + \frac{m}{2} \|x_t^* - x_t\|^2$$
$$\leq f_t(x_t^*) + \lambda_1 D_h(x_t^* || x_{t-1}) + \lambda_2 D_h(x_t^* || v_t). \tag{27}$$

We define the potential function as $\phi(x_t, x_t^*) = (\lambda_1 + \lambda_2) D_h(x_t^* || x_t) + \frac{m}{2} \|x_t^* - x_t\|^2$, and let $\Delta\phi = \phi(x_t, x_t^*) - \phi(x_{t-1}, x_{t-1}^*)$. Applying this notation to inequality (27) and rearranging terms, we obtain

$$H_t + \lambda_1 M_t + \Delta\phi$$
$$\leq (H_t^* + \lambda_2 D_h(x_t^* || v_t)) + \lambda_1 D_h(x_t^* || x_{t-1}) - (\lambda_1 + \lambda_2) D_h(x_{t-1}^* || x_{t-1}) - \frac{m}{2} \|x_{t-1}^* - x_{t-1}\|^2. \tag{28}$$

Using Lemma 1, we get

$$\frac{1}{2\beta} \|\nabla h(x_{t-1}) - \nabla h(x_{t-1}^*)\|_*^2 \leq D_h(x_{t-1}^* || x_{t-1}), \tag{29}$$

and

$$\|\nabla h(x_{t-1}) - \nabla h(x_{t-1}^*)\|_* \leq \beta \|x_{t-1} - x_{t-1}^*\|. \tag{30}$$

Using Lemma 5 and the two above inequalities, we get

$$\lambda_1 D_h(x_t^* || x_{t-1}) - (\lambda_1 + \lambda_2) D_h(x_{t-1}^* || x_{t-1}) - \frac{m}{2} \|x_{t-1}^* - x_{t-1}\|^2$$

$$= \lambda_1 \left( D_h(x_t^* || x_{t-1}) - D_h(x_{t-1}^* || x_{t-1}) \right) - \lambda_2 D_h(x_{t-1}^* || x_{t-1}) - \frac{m}{2} \|x_{t-1}^* - x_{t-1}\|^2 \tag{31a}$$

$$= \lambda_1 D_h(x_t^* || x_{t-1}^*) + \lambda_1 \langle \nabla h(x_{t-1}) - \nabla h(x_{t-1}^*), x_{t-1}^* - x_t^* \rangle$$
$$\quad - \lambda_2 D_h(x_{t-1}^* || x_{t-1}) - \frac{m}{2} \|x_{t-1}^* - x_{t-1}\|^2 \tag{31b}$$

$$\leq \lambda_1 D_h(x_t^* || x_{t-1}^*) + \lambda_1 \|\nabla h(x_{t-1}) - \nabla h(x_{t-1}^*)\|_* \|x_{t-1}^* - x_t^*\|$$
$$\quad - \lambda_2 D_h(x_{t-1}^* || x_{t-1}) - \frac{m}{2} \|x_{t-1}^* - x_{t-1}\|^2 \tag{31c}$$

$$\leq \lambda_1 D_h(x_t^* || x_{t-1}^*) + \frac{\lambda_2 \beta + m}{2\beta^2} \|\nabla h(x_{t-1}) - \nabla h(x_{t-1}^*)\|_*^2 + \frac{\lambda_1^2 \beta^2}{2(\lambda_2 \beta + m)} \|x_{t-1}^* - x_t^*\|^2$$
$$\quad - \lambda_2 D_h(x_{t-1}^* || x_{t-1}) - \frac{m}{2} \|x_{t-1}^* - x_{t-1}\|^2$$

$$= \lambda_1 D_h(x_t^* || x_{t-1}^*) + \frac{\lambda_1^2 \beta^2}{2(\lambda_2 \beta + m)} \|x_{t-1}^* - x_t^*\|^2$$
$$\quad + \left( \frac{\lambda_2}{2\beta} \|\nabla h(x_{t-1}) - \nabla h(x_{t-1}^*)\|_*^2 - \lambda_2 D_h(x_{t-1}^* || x_{t-1}) \right)$$
$$\quad + \left( \frac{m}{2\beta^2} \|\nabla h(x_{t-1}) - \nabla h(x_{t-1}^*)\|_*^2 - \frac{m}{2} \|x_{t-1}^* - x_{t-1}\|^2 \right) \tag{31d}$$

$$\leq \lambda_1 D_h(x_t^* || x_{t-1}^*) + \frac{\lambda_1^2 \beta^2}{2(\lambda_2 \beta + m)} \|x_{t-1}^* - x_t^*\|^2$$

$$\leq \lambda_1 \left( 1 + \frac{\lambda_1 \beta^2}{\alpha(\lambda_2 \beta + m)} \right) D_h(x_t^* || x_{t-1}^*), \tag{31e}$$

where we use Lemma 5 in line (31a); Cauchy-Schwartz inequality in line (31b); the AM-GM inequality in the line (31c); inequalities (29) and (30) in line (31d); and inequality (25) in line (31e).

Substituting inequality (31) into inequality (28), we obtain

$$H_t + \lambda_1 M_t + \Delta\phi \le \left( H_t^* + \lambda_2 D_h(x_t^* || v_t) \right) + \lambda_1 \left( 1 + \frac{\lambda_1 \beta^2}{\alpha(\lambda_2 \beta + m)} \right) M_t^*.$$

Using inequality (25) and the fact that $f_t$ is $m$-strongly convex, we obtain

$$\lambda_2 D_h(x_t^* || v_t) \le \frac{\lambda_2 \beta}{2} \| x_t^* - v_t \|^2 \le \frac{\lambda_2 \beta}{m} H_t^*.$$

Therefore we have

$$H_t + \lambda_1 M_t + \Delta\phi \le (1 + \frac{\lambda_2 \beta}{m}) H_t^* + \lambda_1 \left( 1 + \frac{\lambda_1 \beta^2}{\alpha(\lambda_2 \beta + m)} \right) M_t^*.$$

Since $0 < \lambda_1 \le 1$, we have

$$H_t + M_t + \frac{1}{\lambda_1} \Delta\phi \le \frac{H_t + \lambda_1 M_t + \Delta\phi}{\lambda_1} \le \frac{m + \lambda_2 \beta}{m\lambda_1} H_t^* + \left( 1 + \frac{\beta^2}{\alpha} \cdot \frac{\lambda_1}{\lambda_2 \beta + m} \right) M_t^*.$$

Theorem 4 follows from summing the above inequality over all timesteps $t$.

# E  R-OBD with Squared $\ell_2$ Norm

When $h(x) = \frac{1}{2} \|x\|_2^2$, the Bregman Divergence $D_h(x||y)$ is equal to the squared $\ell_2$ norm $\frac{1}{2} \|x - y\|_2^2$. Hence, setting $h(x) = \frac{1}{2} \|x\|_2^2$ in Algorithm 3 gives us R-OBD in the squared $\ell_2$ setting. In this section, we present a separate proof of Regularized OBD with squared $\ell_2$ norm, in order to remove the assumption that the hitting costs $\{f_t\}$ are differentiable.

**Theorem 7.** *Consider hitting cost functions that are $m$-strongly convex with respect to $\ell_2$ norm and movement costs given by $\frac{1}{2} \|x_t - x_{t-1}\|_2^2$. There exists a choice $\lambda_1, \lambda_2$ such that the competitive ratio of Regularized OBD matches the lower bound proved in Theorem 1, i.e. the competitive ratio is at most $\frac{1}{2} \left( 1 + \sqrt{1 + \frac{4}{m}} \right)$.*

This result follows from the more general bound in Theorem 8 below, which describes the competitive ratio of Algorithm 3 as a function of $\lambda_1, \lambda_2$.

**Theorem 8.** *Consider hitting cost functions that are $m$-strongly convex with respect to $\ell_2$ norm and movement costs given by $\frac{1}{2} \|x_t - x_{t-1}\|_2^2$. Regularized-OBD (Algorithm 3 with $h(x) = \frac{1}{2} \|x\|_2^2$) with parameters $1 \ge \lambda_1 > 0, \lambda_2 \ge 0$ has competitive ratio at most*

$$\max \left( \frac{m + \lambda_2}{\lambda_1} \cdot \frac{1}{m}, 1 + \frac{\lambda_1}{\lambda_2 + m} \right).$$

Notice that Theorem 7 follows immediately by setting $\frac{m + \lambda_2}{\lambda_1} = \frac{m}{2} \left( 1 + \sqrt{1 + \frac{4}{m}} \right)$ in Theorem 8.

Before proving Theorem 8, we first prove a teechnical lemma which gives a lower bound of the value of hitting cost as a function of the distance to the minimizer.

**Lemma 13.** *If $f : \mathcal{X} \to \mathbb{R}$ is a $m$-strongly convex function with respect to some norm $\|\cdot\|$, and $v$ is the minimizer of $f$ (i.e. $v = \arg\min_{x \in \mathcal{X}} f(x)$), then we have $\forall x \in \mathcal{X}$,*

$$f(x) \ge f(v) + \frac{m}{2} \|x - v\|^2.$$

*Proof.* By the definition of $m$-strongly convex, we obtain that $\forall \alpha \in (0, 1)$,

$$f(\alpha x + (1 - \alpha)v) \le \alpha f(x) + (1 - \alpha)f(v) - \frac{m}{2}\alpha(1 - \alpha) \|x - v\|^2. \tag{32}$$

772  Notice that $f(v) \leq f(\alpha x + (1-\alpha)v)$. Combining this with inequality (32), we obtain that $\forall \alpha \in (0,1)$,

$$f(v) \leq \alpha f(x) + (1-\alpha)f(v) - \frac{m}{2}\alpha(1-\alpha)\|x-v\|^2.$$

773  Rearranging the terms, we observe that $\forall \alpha \in (0,1)$,

$$f(x) \geq f(v) + \frac{m}{2}(1-\alpha)\|x-v\|^2.$$

774  Therefore

$$f(x) \geq \lim_{\alpha \to 0^+}\left(f(v) + \frac{m}{2}(1-\alpha)\|x-v\|^2\right) = f(v) + \frac{m}{2}\|x-v\|^2.$$

775  $\qquad\qquad\qquad\qquad\qquad\qquad\qquad\qquad\qquad\qquad\qquad\qquad\qquad\qquad\qquad\qquad\qquad\qquad$ $\square$

776  Now we return to the proof of Theorem 8.

777  *Proof of Theorem 8.* In the proof, we use the property of strongly convex to derive an inequality in
778  the form of $H_t + M_t + \Delta\phi \leq C(H_t^* + M_t^*)$, where $\Delta\phi$ is the change in potential and $C$ is an upper
779  bound for the competitive ratio.

780  Throughout the proof, we use $\|\cdot\|$ to denote $\ell_2$ norm.

781  Notice that when $h(x) = \frac{1}{2}\|x\|^2$, the update rule in Algorithm 3 is:

$$x_t = \arg\min_x f_t(x) + \frac{\lambda_1}{2}\|x - x_{t-1}\|^2 + \frac{\lambda_2}{2}\|x - v_t\|^2.$$

782  For convenience, we define

$$F_t(x) = f_t(x) + \frac{\lambda_1}{2}\|x - x_{t-1}\|^2 + \frac{\lambda_2}{2}\|x - v_t\|^2.$$

783  Since $f_t(x)$ is $m$-strongly convex, $\frac{\lambda_1}{2}\|x - x_{t-1}\|^2$ is $\lambda_1$-strongly convex, and $\frac{\lambda_2}{2}\|x - v_t\|^2$ is $\lambda_2$-
784  strongly convex, $F_t(x)$ is $(m + \lambda_1 + \lambda_2)$−strongly convex. Since $x_t = \arg\min_x F_t(x)$, by Lemma
785  13, we obtain

$$F_t(x_t^*) \geq F_t(x_t) + \frac{m + \lambda_1 + \lambda_2}{2}\|x_t^* - x_t\|^2,$$

786  which implies

$$
\begin{aligned}
&H_t + \lambda_1 M_t + \frac{m + \lambda_1 + \lambda_2}{2}\|x_t^* - x_t\|^2 \\
&\leq H_t + \lambda_1 M_t + \frac{\lambda_2}{2}\|x - v_t\|^2 + \frac{m + \lambda_1 + \lambda_2}{2}\|x_t^* - x_t\|^2 \\
&\leq H_t^* + \frac{\lambda_1}{2}\|x_t^* - x_{t-1}\|^2 + \frac{\lambda_2}{2}\|x_t^* - v_t\|^2.
\end{aligned}
\tag{33}
$$

787  We define the potential function as $\phi(x_t, x_t^*) = \frac{m + \lambda_1 + \lambda_2}{2}\|x_t^* - x_t\|^2$ and $\Delta\phi = \phi(x_t, x_t^*) -$
788  $\phi(x_{t-1}, x_{t-1}^*)$. We then can rewrite inequality (33) as

$$H_t + \lambda_1 M_t + \Delta\phi \leq \left(H_t^* + \frac{\lambda_2}{2}\|x_t^* - v_t\|^2\right) + \frac{\lambda_1}{2}\|x_t^* - x_{t-1}\|^2 - \frac{m + \lambda_1 + \lambda_2}{2}\|x_{t-1}^* - x_{t-1}\|^2.$$

$$\tag{34}$$

Additionally

$$\frac{\lambda_1}{2}\left\|x_t^* - x_{t-1}\right\|^2 - \frac{m + \lambda_1 + \lambda_2}{2}\left\|x_{t-1}^* - x_{t-1}\right\|^2$$

$$\leq \frac{\lambda_1}{2}\left(\left\|x_t^* - x_{t-1}^*\right\| + \left\|x_{t-1}^* - x_{t-1}\right\|\right)^2 - \frac{m + \lambda_1 + \lambda_2}{2}\left\|x_{t-1}^* - x_{t-1}\right\|^2 \tag{35a}$$

$$= \frac{\lambda_1}{2}\left\|x_t^* - x_{t-1}^*\right\|^2 + \lambda_1\left\|x_t^* - x_{t-1}^*\right\| \cdot \left\|x_{t-1}^* - x_{t-1}\right\| - \frac{m + \lambda_2}{2}\left\|x_{t-1}^* - x_{t-1}\right\|^2$$

$$\leq \frac{\lambda_1}{2}\left\|x_t^* - x_{t-1}^*\right\|^2 + \frac{\lambda_1^2}{2(m + \lambda_2)}\left\|x_t^* - x_{t-1}^*\right\|^2 + \frac{m + \lambda_2}{2}\left\|x_{t-1}^* - x_{t-1}\right\|^2$$

$$- \frac{m + \lambda_2}{2}\left\|x_{t-1}^* - x_{t-1}\right\|^2 \tag{35b}$$

$$= \frac{\lambda_1(\lambda_1 + \lambda_2 + m)}{2(\lambda_2 + m)}\left\|x_{t-1}^* - x_{t-1}^*\right\|^2$$

$$= \lambda_1\left(1 + \frac{\lambda_1}{\lambda_2 + m}\right)M_t^*,$$

where we apply the triangle inequality in line (35a) and AM-GM in line (35b).

Combining inequalities (34) and (35), we obtain

$$H_t + \lambda_1 M_t + \Delta\phi \leq \left(H_t^* + \frac{\lambda_2}{2}\left\|x_t^* - v_t\right\|^2\right) + \lambda_1\left(1 + \frac{\lambda_1}{\lambda_2 + m}\right)M_t^*. \tag{36}$$

And since $f_t(x)$ is $m$-strongly convex, we have

$$\frac{\lambda_2}{2}\left\|x_t^* - v_t\right\|^2 \leq \frac{\lambda_2}{m}H_t^*.$$

Substituting the above identity into inequality (36) yields

$$H_t + \lambda_1 M_t + \Delta\phi \leq \frac{m + \lambda_2}{m}H_t^* + \lambda_1\left(1 + \frac{\lambda_1}{m + \lambda_2}\right)M_t^*. \tag{37}$$

Using inequality (37), we obtain

$$H_t + M_t + \frac{1}{\lambda_1}\Delta\phi \leq \frac{H_t + \lambda_1 M_t + \Delta\phi}{\lambda_1} \leq \frac{m + \lambda_2}{\lambda_1 m}H_t^* + \left(1 + \frac{\lambda_1}{m + \lambda_2}\right)M_t^*.$$

Theorem 8 follows from summing the above inequality over all timesteps $t$. $\qquad\square$

# F  Proof of Theorem 5

In this proof, we construct counterexamples for two separate cases, based on whether $\lambda_1$ is larger or smaller than $m$. Recall that $\lambda_2 = 0$ throughout the proof.

**Case 1: $\lambda_1 > m$**

In this case, we show the competitive ratio can be unbounded by proposing a series of identical hitting cost functions on the real number line. We construct a hitting cost function $f$ with minimizer $v$ so that there exists a fixed point $K \neq v$ (i.e. when $x_{t-1} = K$ and $f_t = f$, the algorithm selects $x_t = x_{t-1}$). Since R-OBD is independent of timestep $t$, we can propose $f_t = f$ for $t = 1, 2, \cdots, T$ and let $x_0 = K$. In this scenario, the total cost of R-OBD grows linearly in $T$. However, by choosing $x_1 = x_2 = \cdots = x_T = v$, the total cost incurred by the offline adversary is a constant. Therefore the competitive ratio of R-OBD will be unbounded.

Specifically, consider the hitting cost function

$$f(x) = \begin{cases} \frac{m}{2}\left(1 - (x+1)^2\right) & -1 \leq x \leq 0 \\ \frac{m}{2}x^2 & \text{otherwise} \end{cases}.$$

Suppose $x_{t-1} = -1$, then R-OBD will choose $x_t$ such that

$$x_t = \arg\min_x f(x) + \frac{\lambda_1}{2}(x+1)^2.$$

Notice that

$$f(x) + \frac{\lambda_1}{2}(x+1)^2 = \begin{cases} \frac{m}{2} + \frac{\lambda_1 - m}{2}(x+1)^2 & -1 \le x \le 0 \\ \frac{m}{2}x^2 + \frac{\lambda_1}{2}(x+1)^2 & \text{otherwise} \end{cases}.$$

Since $\lambda_1 > m$, we see that the quantity above is $\geq \frac{m}{2}$ for all real $x$, where equality only holds when $x = -1$. It follows that $x_t = x_{t-1} = -1 \neq 0 = v$. Thus $K = -1$ is a fixed point satisfying the requirements described as above.

**Case 2:** $\lambda_1 \le m$

We consider a situation such that the R-OBD algorithm moves far away from the starting point, incurring significant movement cost, whereas the offline adversary could pay relatively little cost by staying at the starting point. More specifically, suppose the starting point $x_0 = 0$ and the first hitting cost function is $f_1(x) = \frac{m}{2}(1-x)^2$. Consider an adversary which chooses $x_0 = x_1 = \cdots = x_T$. The cost incurred by the adversary is

$$cost(ADV) = \frac{m}{2}.$$

Using the update rule, the R-OBD algorithm chooses

$$x_1 = \arg\min_x \frac{m}{2}(1-x)^2 + \frac{\lambda_1}{2}x^2 = \frac{m}{m+\lambda_1} \geq \frac{1}{2}.$$

The movement cost incurred by R-OBD is at least

$$cost(ALG) \geq M_1 = \frac{1}{2}x_1^2 \geq \frac{1}{8}.$$

Thus the competitive ratio is at least

$$\frac{cost(ALG)}{cost(ADV)} \geq \frac{1}{4m}.$$

Theorem 5 follows from combining these two cases.

# G Proof of Theorem 6

Let $\{x_t^L\}$ be the sequence of points achieving the $L$-constrained offline optimal . We first prove an upper bound on the difference of hitting costs $f_t(x_t) - f_t(x_t^L)$, and then use this bound to prove a $O\left(G\sqrt{TL}\right)$ upper bound on the regret $\sum_{t=1}^{T}\left(f_t(x_t) - f_t(x_t^L) + c(x_t, x_{t-1})\right) - \sum_{t=1}^{T} c(x_t^L, x_{t-1}^L)$.

Since the function $f_t(x) + \lambda_1 D_h(x||x_{t-1}) + \lambda_2 D_h(x||v_t)$ is strongly convex, it has a unique minimizer, at which point the gradient vanishes. This is the point $x_t$ which Algorithm 3 picks in round $t$. We can rearrange the vanishing gradient condition to obtain

$$\nabla f_t(x_t) = \lambda_1 \left(\nabla h(x_{t-1}) - \nabla h(x_t)\right) + \lambda_2 \left(\nabla h(v_t) - \nabla h(x_t)\right).$$

Therefore by Lemma 5, we have

$$
\begin{aligned}
\langle \nabla f_t(x_t), x_t - x_t^L \rangle &= \lambda_1 \langle \nabla h(x_{t-1}) - \nabla h(x_t), x_t - x_t^L \rangle + \lambda_2 \langle \nabla h(v_t) - \nabla h(x_t), x_t - x_t^L \rangle \\
&= \lambda_1 \left(D_h(x_t^L||x_{t-1}) - D_h(x_t^L||x_t) - D_h(x_t||x_{t-1})\right) \\
&\quad + \lambda_2 \left(D_h(x_t^L||v_t) - D_h(x_t^L||x_t) - D_h(x_t||v_t)\right).
\end{aligned}
$$
(38)

Recall that $h$ is $\alpha-$strongly convex and $\beta-$strongly smooth with respect to the norm $\|\cdot\|$, hence

$$\forall x, y, \frac{\alpha}{2}\|x-y\|^2 \le D_h(x||y) \le \frac{\beta}{2}\|x-y\|^2.$$
(39)

Therefore

$$D_h(x_t^L||v_t) - D_h(x_t^L||x_t) - D_h(x_t||v_t) \leq D_h(x_t^L||v_t) \leq \frac{\beta}{2}\left\|x_t^L - v_t\right\|^2 \leq \frac{\beta D^2}{2}.$$

In light of equation (38), we obtain

$$\langle\nabla f_t(x_t), x_t - x_t^L\rangle \leq \lambda_1\left(D_h(x_t^L||x_{t-1}) - D_h(x_t^L||x_t) - D_h(x_t||x_{t-1})\right) + \frac{\beta D^2}{2}\cdot\lambda_2. \quad (40)$$

Let $q > 0$ be a parameter which we will pick later. For all $q > 0$, it holds that

$$f_t(x_t) - f_t(x_t^L)$$

$$\leq \langle\nabla f_t(x_t), x_t - x_t^L\rangle - \frac{m}{2}\left\|x_t - x_t^L\right\|^2 \qquad (41a)$$

$$\leq \lambda_1\left(D_h(x_t^L||x_{t-1}) - D_h(x_t^L||x_t) - D_h(x_t||x_{t-1})\right) - \frac{m}{2}\left\|x_t - x_t^L\right\|^2 + \frac{\beta D^2}{2}\cdot\lambda_2 \qquad (41b)$$

$$= (\lambda_1 + q)\left(D_h(x_t^L||x_{t-1}) - D_h(x_t^L||x_t)\right) - \lambda_1 D_h(x_t||x_{t-1})$$

$$\quad - \left(qD_h(x_t^L||x_{t-1}) - qD_h(x_t^L||x_t) + \frac{m}{2}\left\|x_t - x_t^L\right\|^2\right)$$

$$\quad + \frac{\beta D^2}{2}\cdot\lambda_2.$$

where we apply strong convexity in line (41a), and equation (40) in line (41b). Using Lemma 5, we obtain

$$qD_h(x_t^L||x_{t-1}) - qD_h(x_t^L||x_t) + \frac{m}{2}\left\|x_t - x_t^L\right\|^2$$

$$= qD_h(x_t||x_{t-1}) + q\langle\nabla h(x_{t-1}) - \nabla h(x_t), x_t - x_t^L\rangle + \frac{m}{2}\left\|x_t - x_t^L\right\|^2$$

$$\geq qD_h(x_t||x_{t-1}) - q\left\|\nabla h(x_{t-1}) - \nabla h(x_t)\right\|_*\left\|x_t - x_t^L\right\| + \frac{m}{2}\left\|x_t - x_t^L\right\|^2 \qquad (42a)$$

$$\geq qD_h(x_t||x_{t-1}) - \left(\frac{q^2}{2m}\left\|\nabla h(x_{t-1}) - \nabla h(x_t)\right\|_*^2 + \frac{m}{2}\left\|x_t - x_t^L\right\|^2\right) + \frac{m}{2}\left\|x_t - x_t^L\right\|^2 \quad (42b)$$

$$= qD_h(x_t||x_{t-1}) - \frac{q^2}{2m}\left\|\nabla h(x_{t-1}) - \nabla h(x_t)\right\|_*^2$$

$$\geq qD_h(x_t||x_{t-1}) - \frac{\beta q^2}{m}D_h(x_t||x_{t-1}) \qquad (42c)$$

$$= \left(q - \frac{\beta q^2}{m}\right)D_h(x_t||x_{t-1}),$$

where we apply the Cauchy-Schwartz inequality in line (42a), the AM-GM inequality in line (42b), and Lemma 1 in line (42c).

In order to maximize the coefficient $\left(q - \frac{\beta q^2}{m}\right)$, we set $q = \frac{m}{2\beta}$. By substituting inequality (42) into inequality (41), we obtain

$$f_t(x_t) - f_t(x_t^L)$$

$$\leq \left(\lambda_1 + \frac{m}{2\beta}\right)\left(D_h(x_t^L||x_{t-1}) - D_h(x_t^L||x_t)\right) - \left(\lambda_1 + \frac{m}{4\beta}\right)D_h(x_t||x_{t-1}) + \frac{\beta D^2}{2}\cdot\lambda_2. \qquad (43)$$

Using the condition $\lambda_1 + \frac{m}{4\beta} \geq 1$, we observe that

$$f_t(x_t) - f_t(x_t^L) + D_h(x_t||x_{t-1})\left(\lambda_1 + \frac{m}{2\beta}\right)\left(D_h(x_t^L||x_{t-1}) - D_h(x_t^L||x_t)\right) + \frac{\beta D^2}{2}\cdot\lambda_2. \quad (44)$$

Notice that

$$\sum_{t=1}^{T}\left\|x_t^L - x_{t+1}^L\right\| \leq \sqrt{T\left(\sum_{t=1}^{T}\left\|x_t^L - x_{t+1}^L\right\|^2\right)} \leq \sqrt{T\left(\sum_{t=1}^{T}\frac{2D_h(x_{t+1}^L||x_t^L)}{\alpha}\right)} \leq \sqrt{\frac{2TL}{\alpha}}.$$

$$(45)$$

where we use the generalized mean inequality in the first step and $\alpha$-strong convexity of $h$ in the second step (cf. equation (39)). By Lemma 6, we can give the following upper bound:

$$\sum_{t=1}^{T} D_h(x_t^L || x_{t-1}) - D_h(x_t^L || x_t)$$

$$= \sum_{t=1}^{T} \left( D_h(0 || x_{t-1}) - D_h(0 || x_t) + \langle \nabla h(x_t) - \nabla h(x_{t-1}), x_t^L \rangle \right)$$

$$= D_h(0 || x_0) - D_h(0 || x_T) + \sum_{t=1}^{T-1} \langle \nabla h(x_t), x_t^L - x_{t+1}^L \rangle - \langle \nabla h(x_0), x_1^L \rangle + \langle \nabla h(x_T), x_T^L \rangle$$

$$\leq \sum_{t=1}^{T} \langle \nabla h(x_t), x_t^L - x_{t+1}^L \rangle \tag{46a}$$

$$\leq \sum_{t=1}^{T} \|\nabla h(x_t)\|_* \left\| x_t^L - x_{t+1}^L \right\| \tag{46b}$$

$$\leq G \sum_{t=1}^{T} \left\| x_t^L - x_{t+1}^L \right\|$$

$$\leq G \sqrt{\frac{2TL}{\alpha}}, \tag{46c}$$

where we use the facts $x_0 = x_0^L = x_{T+1}^L = 0, \nabla h(0) = 0$ in line (46a), the Cauchy-Schwartz inequality in line (46b), and inequality (45) in line (46c).

Therefore we obtain

$$cost(OBD) - cost(OPT(L))$$

$$= \sum_{t=1}^{T} \left( f_t(x_t) + D_h(x_t || x_{t-1}) \right) - \left( f_t(x_t^L) + D_h(x_t^L || x_{t-1}^L) \right) \tag{47a}$$

$$\leq \left( \sum_{t=1}^{T} f_t(x_t) - f_t(x_t^L) + D_h(x_t || x_{t-1}) \right) - L$$

$$\leq \left( \lambda_1 + \frac{m}{2\beta} \right) G \sqrt{\frac{2TL}{\alpha}} + T \cdot \frac{\beta D^2}{2} \cdot \lambda_2 - L, \tag{47b}$$

where we use the definition of $OPT(L)$ in line (47a); inequalities (44) and (46) in line (47b).

Since by assumption we have $G < \infty$, $\lambda_2 = \eta(T, L, D, G) \leq \frac{KG}{D^2} \cdot \sqrt{\frac{L}{T}}$ for some constant $K$, by inequality (47), we obtain

$$cost(OBD) - cost(OPT(L)) = O(G\sqrt{TL}),$$

which completes the proof.