[Reviews · NeurIPS 2019]

Reviewer 1



I have a few questions and would appreciate if the authors can answer them in the rebuttal and / or the final version of the paper : 1. The paper deals with the movement cost ||x_t - x_{t-1} ||^2 as compared to ||x_t - x_{t-1} || (which one might expect in standard applications). What happens for the latter case ? Do the upper bounds / lower bounds and the new algorithm extend to that setting ? 2. Are there any lower bounds or impossibility results for regret when L is large (in the definition of L-constrained dynamic regret, Theorem 6) ? 3. Why is the \gamma parameter not tunable in the lower bound of Theorem 2 for OBD. 4. Suppose f_t’s are G lipschitz. What can be said then ? It seems that the current lower bound will not work if the hitting losses are forced to have bounded gradients. Is online balanced descent optimal then ? 5. Is the regret still bounded when f_t is stochastic and not known before making the decision x_t. It seems that the competitive ratio is hard to bound, but regret may still be controllable. Writing Style: The paper is overall well written. There are some minor typos (e.g. lemma-1 in appendix; line 281, etc), but I am sure they will be corrected in the final version of the paper. ============ Post Author Feedback ============ I want to thank the authors for giving me a detailed response to my review and, sharing connections of their work to the existing literature. I am increasing my score by 1 point (from 7 to 8) to acknowledge my satisfaction. Best !

Reviewer 2



--------- post rebuttal comments: After carefully reading the comments from the other two reviewers and the responses provided by the authors on all the comments, I am willing to upgrade my score from 6 to 7. My concern about the paper being "highly specialized" remains, but it will be of interest to some within NeurIPS. --------- a) summary of the content: In this paper, the authors consider an online convex optimization variant where an online learner plays a series of rounds against an adaptive adversary. In each round $t$, the adversary picks a convex cost function $f_t$ and reveals it to the learner. The learner chooses a point $x_t$ and incurs a hitting cost $f_t(x_t)$ and a movement cost $c(x_{t-1},x_t)$ associated with the change of point over the previous round. The objective of the learner is to minimize the sum over $T$ rounds of per-round hitting costs and movement costs, and design online algorithms that perform well against the adversary, as measured by competitive ratios. Concentrating on the case of the $\ell_2$-squared norm $c(x_{t-1},x_t)=\frac{1}{2} \|x_t-x_{t-1} \|_2^2$ and $m$-strongly convex functions $f_t$, the authors show a lower bound of $\Omega(m^{-1/2})$ on any online algorithms, show that an existing algorithm called Online Balanced Decent (OBD) has a lower bound of $\Omega(m^{-2/3})$, and propose two optimal variants, Greedy OBD (G-OBD) and Regularized OBD (R-OBD) with an $O(m^{-1/2})$ competitive ratio. They also show that assuming that under additional smoothness for $f_t$, R-OBD can simultaneously achieve a constant, dimension-free competitive ratio and sublinear ($L$-constrained dynamic) regret. b) strengths and weaknesses of the submission. * originality: This is a highly specialized contribution building up novel results on two main fronts: The derivation of the lower bound on the competitive ratio of any online algorithm and the introduction of two variants of an existing algorithm so as to meet this lower bound. Most of the proofs and techniques are natural and not surprising. In my view the main contribution is the introduction of the regularized version which brings a different, and arguably more modern interpretation, about the conditions under which these online algorithms perform well in these adversarial settings. * quality: The technical content of the paper is sound and rigorous * clarity: The paper is in general very well-written, and should be easy to follow for expert readers. * significance: As mentioned above this is a very specialized paper likely to interest some experts in the online convex optimization communities. Although narrow in scope, it contains interesting theoretical results advancing the state of the art in dealing with these specific problems. * minor details/comments: - p.1, line 6-7: I would rewrite the sentence to simply express that the lower bound is $\Omega(m^{-1/2})$ \- p.3, line 141: cost an algorithm => cost of an algorithm \- p.4, Algorithm 1, step 3: mention somewhere that this is the projection operator (not every reader will be familiar with this notation \- p.5, Theorem 2: remind the reader that the $\gamma$ in the statement is the parameter of OBD as defined in Algorithm 1 \- p.8, line 314: why surprisingly?

Reviewer 3



The paper derives novel bounds for Smoothed Online Convex Optimization (SOCO) and a particular state of the art algorithm on Online Balanced Descent (OBD), one in , showing its suboptimality, for the setup with (m-)strongly convex hitting costs and the squared l2-norm as movement costs. It also introduces two novel variants of OBD: G-OBD and R-OBD, and shows their optimality in terms of the strong convexity parameter. The paper takes big steps towards understanding Smoothed Online Convex Optimization (SOCO) by addressing and solving various unresolved problems in it: - It provides the first non-trivial lower bounds on SOCO with (m-)strongly convex hitting costs and the squared l2-norm as movement costs. The bound grows as \Omega(m^{-1/2}) as m goes to 0. - It introduces two variants of a state of the art algorithm, Online Balanced Descent (OBD): G-OBD and R-OBD. R-OBD matches the exact lower bound on the above setup, and thus optimal in terms of the strong convexity parameter. G-OBD, on the other hand, has slightly less competitive ratio, O(m^{-1/2}), but on a broader class of problems. - It proves a Omega(m^{-2/3}) lower bound on the competitive ratio of Online Balanced Descent (OBD), a state of the art algorithm in OBD. This thus shows its suboptimality in terms of the strong convexity parameter. - The presented bounds imply that R-OBD can achieve dimension-free competitive ratio and sublinear regret simultaneously for problems with m-strongly convex hitting costs and the squared l2-norm as movement costs. (For linear hitting and movement costs, this was known to be impossible.) The obtained results are strong contributions that improve our understanding of the topic and potentially help solving further problems in the area. The subtleties of the results and the problems are essential, which automatically renders the paper hard to approach; however the authors did a really good job in highlighting the main details and guiding the reader through several layers of the problem. The summary of the main ideas were also very efficient in conveying the main challenges and ideas. Overall, it was an enjoyable read. Minor remarks: line 179: "costs costs" -> "costs" Lemma 1: please quantify \beta. Lemma 4: this is a well-known property of convex conjugates

[Author Response · NeurIPS 2019]

Note that the numbered citations refer to references in the submitted paper. The remaining citations are listed at the end of the page.

**Reviewer 1.** We thank you for your detailed commentary. We respond to your questions in order:

1. Within the literature on SOCO, both movement costs that are "squared" and "linear" have been studied. We have focused on the "squared" case here due to its applications to control, e.g., LQR control, which was first shown in [28] and later extended in [22]. The squared case also applies to online regression problems and economic dispatch in power systems. The ideas in the submitted paper also apply to the "linear" case. We have new results showing that G-OBD provides an order optimal competitive ratio among a class of memoryless algorithms in the "linear" case, as defined in [18], but we have yet to succeed in analyzing R-OBD in the "linear" setting. Understanding the "linear" case in more detail is a topic of our ongoing research.

2. Theorem 6 in our submission does not make additional assumptions on the scale of $L$. However, we have not obtained a lower bound on dynamic regret in the same problem setting thus far. In [28] a lower bound was obtained for a different notion of dynamic regret defined by the path length of minimizers. Unfortunately, the two definitions are not comparable.

3. The point of Theorem 2 is to show that OBD has suboptimal competitive ratio for any choice of the balance parameter $\gamma$. In particular, without some new insight, we cannot create an optimal version of OBD simply by adjusting $\gamma$. This observation shows the need to extend OBD, which we do in the remainder of the paper.

4. While we have not investigated $\ell$-Lipshitz functions, the setting of well-conditioned cost functions is of particular interest and we have some additional results in that case. Specifically, when the condition number of the hitting cost functions is $k$ the competitive ratio of R-OBD is bounded above by $1 + k$. However, the lower bound in the same setting is still a topic of our ongoing research.

5. Some stochastic models have been considered in related work [17], but we have not analyzed R-OBD in those situations. This is a topic of ongoing research.

**Reviewer 2.** We thank you for your detailed commentary. You have suggested that the problem we consider is "highly specialized." We feel strongly that SOCO is an important problem in the online learning space, and one with stunningly broad range of applications including data center capacity provisioning, demand response, speech animation, video streaming, network function virtualization, and more [24, 26, 27, 29][SLJ19]. In many of these applications, approaches based on SOCO algorithms have been deployed in production systems, e.g., within HP, Disney, Microsoft, Akaimai, Southern California Edison, and Google (among others). An example of a particularly exciting recent connection is that SOCO was used to analyze and derive policies for LQR control problems [22], which is a core problem in the community. Another important connection is to the problem of Online Body Chasing, which has led to considerable interest in the algorithms community [Sel19, AGGT19]. Because of these applications, this classical problem has received a surge of interest across the networking, theoretical computer science, machine learning, and energy communities, with multiple papers in conferences such as Sigmetrics, FOCS/STOC, e-Energy, and COLT appearing each year.

**Reviewer 3.** We thank you for your detailed commentary, and for pointing out some typos and redundancies which we will correct in the final version of our paper. You mentioned a slight concern with the significance of the problem setting. We hope that our response to reviewer 2 eases this concern.

# References

[AGGT19] CJ Argue, Anupam Gupta, Guru Guruganesh, and Ziye Tang. Chasing convex bodies with linear competitive ratio. *arXiv preprint arXiv:1905.11877*, 2019.

[Sel19] Mark Sellke. Chasing convex bodies optimally. *arXiv preprint arXiv:1905.11968*, 2019.

[SLJ19] Ming Shi, Xiaojun Lin, and Lei Jiao. On the value of look-ahead in competitive online convex optimization. *Proceedings of the ACM on Measurement and Analysis of Computing Systems*, 3(2):22, 2019.


[Meta-Review · NeurIPS 2019]

This is a solid theory contribution. The reviewers mention the specialized nature of the problem setting. But given the strength of the results and the presentation clarity lauded by all reviewers, I think the paper should be awarded a spotlight presentation.